# Genomic characterization of cervical lymph node metastases in papillary thyroid carcinoma following the Chornobyl accident

Lindsay M. Morton [1,16] ✉, Olivia W. Lee [2,16], Danielle M. Karyadi[2,16], Tetiana I. Bogdanova[3,16], Chip Stewart[4], Stephen W. Hartley [2], Charles E. Breeze[5], Sara J. Schonfeld [1], Elizabeth K. Cahoon [1], Vladimir Drozdovitch [1], Sergii Masiuk [6], Mykola Chepurny[6], Liudmyla Yu Zurnadzhy[3], Jieqiong Dai[7], Marko Krznaric[8], Meredith Yeager [7], Amy Hutchinson[7], Belynda D. Hicks [7], Casey L. Dagnall [7], Mia K. Steinberg[7], Kristine Jones[7], Komal Jain[7], Ben Jordan[7], Mitchell J. Machiela[9], Eric T. Dawson[2,10], Vibha Vij[1], Julie M. Gastier-Foster[11,12], Jay Bowen [11], Kiyohiko Mabuchi[1], Maureen Hatch[1], Amy Berrington de Gonzalez[1], Gad Getz [4,13,14], Mykola D. Tronko[15,17], Gerry A. Thomas[8,17] & Stephen J. Chanock [2,17] ✉

Childhood radioactive iodine exposure from the Chornobyl accident increased papillary thyroid carcinoma (PTC) risk. While cervical lymph node metastases (cLNM) are well-recognized in pediatric PTC, the PTC metastatic process and potential radiation association are poorly understood. Here, we analyze cLNM occurrence among 428 PTC with genomic landscape analyses and known drivers ([131]I-exposed = 349, unexposed = 79; mean age = 27.9 years). We show that cLNM are more frequent in PTC with fusion (55%) versus mutation (30%) drivers, although the proportion varies by specific driver gene (*RET*-fusion = 71%, *BRAF*-mutation = 38%, *RAS*-mutation = 5%). cLNM frequency is not associated with other characteristics, including radiation dose. cLNM molecular profiling (N = 47) demonstrates 100% driver concordance with matched primary PTCs and highly concordant mutational spectra. Transcriptome analysis reveals 17 differentially expressed genes, particularly in the *HOXC* cluster and *BRINP3*; the strongest differentially expressed microRNA also is near *HOXC10*. Our findings underscore the critical role of driver alterations and provide promising candidates for elucidating the biological underpinnings of PTC cLNM.

Thyroid cancer, the most frequently diagnosed malignancy among adolescents and young adults, has a very good prognosis, with 5-year relative survival rates exceeding 99%[1,2]. The most common form, papillary thyroid carcinoma (PTC), is a well-differentiated tumor that accounts for approximately 85% of thyroid cancers, is more common in women, and is typically managed with surgery (lobectomy and/or total thyroidectomy), post-surgery radioactive iodine ([131]I) ablation, and/or systemic therapy[3,4]. However, clinical uncertainties remain regarding optimal treatment approaches, in part because an estimated 20–50% of patients have cervical lymph node metastases (cLNM) at diagnosis, only a fraction of which are identified through pre-operative imaging[4]. Patient and clinical factors that reportedly correlate with

cLNM and impact surgical decisions include younger age at diagnosis, male sex, increased tumor size, extrathyroidal extension, multifocality, location in the upper pole of the thyroid, and history of exposure to ionizing radiation[4,5].

In the last decade, advances in molecular characterization of PTC have demonstrated promise for providing new insights into patient risk stratification that could inform therapeutic decision-making, mainly based on targeted profiling of somatic alterations. Tumor aggressiveness and occurrence of cLNM and distant metastases have been associated with gene alterations, including *BRAF*[V600E], *TERT* promoter, or *TP53* mutations; *RET* fusions; and 22q loss[4,6–13]. However, published results are inconsistent, possibly due to small sample sizes and ascertainment biases; heterogeneity in patient characteristics, particularly age, which is associated with specific molecular alterations[14–17]; and variability in the molecular characteristics assessed in different studies. A recent report suggested that the relationship between thyroid tumor aggressiveness and molecular characteristics could depend on patient characteristics such as age; specifically, the increased frequency of cLNM and distant metastases in thyroid tumors with *RET/NTRK* fusions versus *BRAF* mutations was pronounced in a series of pediatric patients but much less so in older patients from The Cancer Genome Atlas (TCGA)[14,17].

We recently reported a comprehensive molecular characterization of fresh-frozen primary tumor (PT) samples from 440 histologically confirmed PTC (mean age at diagnosis = 28.0 years, range 10.0–45.6) ascertained through the Chornobyl Tissue Bank as part of a study of the consequences of the Chornobyl nuclear power plant accident[18–20]. [131]I-containing fallout from the accident was deposited in the surrounding environment and has been associated with increased thyroid cancer risk, particularly among children who consumed contaminated leafy greens or dairy products from grazing cows[21]. Previous analyses of patients with PTC following the Chornobyl accident have suggested associations between [131]I dose and tumor invasiveness (extrathyroidal extension, lymphatic/vascular invasion, and regional or distant metastases)[22–26], which were possibly restricted to *BRAF*[V600E]-negative tumors[27]. However, the sample sizes were limited and lacked the combination of comprehensive molecular and clinical data.

Here, we expand our prior study of 440 adolescents and young adults with PTC[18] with additional detailed clinical data to identify molecular and clinical predictors of cLNM occurrence, using both multivariable modeling approaches and stratification to disentangle the independent effects of patient, clinical, epidemiologic, and molecular characteristics. We further conduct a comprehensive genomic landscape analysis of 47 cLNM samples by profiling genomic, transcriptomic, and epigenomic characteristics in comparison to matched PT samples (Supplementary Data 1). Our findings provide insights into the molecular processes underlying the development of metastatic PTC and underscore the importance of the specific driver alteration, almost exclusively drawn from the mitogen-activated protein kinase (MAPK) pathway. In addition, we did not confirm the effect of environmental radiation on the occurrence of cLNM.

## Results

### Patients and clinical predictors of cLNM occurrence

The study population included 440 fresh-frozen, pre-treatment primary PTC tumors with high-quality whole genome sequencing (WGS, mean tumor sequencing depth = 89X) and/or mRNA sequencing (mRNA-seq) (374 both, 57 mRNA-seq only, nine WGS only)[18]. WGS and mRNA-seq data were complemented with single nucleotide polymorphism (SNP) microarray genotyping, relative telomere length quantification, DNA methylation profiling, and microRNA (miRNA)-seq as allowed based on biospecimen availability. Within the series of 440 tumors that were histologically confirmed by a panel of pathology experts[19,20], 359 occurred in individuals with well-quantified [131]I exposure before adulthood (≤18 years old) from the Chornobyl

accident[28–30], and 81 occurred in individuals from the same regions in Ukraine who were born >9 months after the accident and thus considered [131]I-unexposed (Table S1). Among the [131]I-exposed individuals, mean estimated radiation dose was 247 mGy (range: 11–8800), 74.9% of patients were female, and the mean age at PTC diagnosis was 29.7 years (range: 13.4–45.6). [131]I-unexposed individuals similarly were predominantly female (81.5%) but tended to be younger at PTC diagnosis (mean: 20.7 years; range: 10.0–29.1).

The study was based at one central tertiary center, namely the Institute for Endocrinology and Metabolism in Kyiv, Ukraine, where most individuals (*N* = 365, 83.0%) underwent total thyroidectomy (Table S1). Nearly half of the tumors (*N* = 206, 46.8%) were classified as pathologic T1, 71 (16.1%) T2, and 163 (37.0%) T3 according to the 7th edition of TNM staging (Fig. S1 and Tables S1, S2)[31]. Of the 163 tumors classified as T3, 106 tumors of any size had evidence of minimal extrathyroidal extension in the fat and connective tissue, 24 had evidence of extrathyroidal extension in the muscle, and 33 had no evidence of extrathyroidal extension but were sized >4 cm. Multifocality was recorded for 78 (17.7%) tumors. Nearly two-thirds (*N* = 262, 64.1%) of the primary lesions were ≤2 cm and only 36 (8.2%) were >4 cm. Metastases were reported at the time of diagnosis for 179 (40.7%) individuals, including 164 (37.3%) with cLNM only (N1M0), 14 (3.2%) with both cLNM and distant metastases (N1M1), and 1 (0.2%) with distant metastases only (N0M1; this individual was excluded from further analyses of cLNM occurrence). Among patients with cLNM, approximately half were N1a (*N* = 87, 48.9%) and half N1b (*N* = 91, 51.1%).

### Molecular predictors of cLNM occurrence

The molecular characteristics of the 440 PT samples included in this study have been described in detail previously[18] and are provided in Supplementary Data 2. Briefly, a single driver was designated for 429 (97.5%) PT, over half (*N* = 253, 59.0%) of which were mutations (simple somatic variants [SSVs]) that occurred most frequently in *BRAF* (*N* = 194, 45.2%) or *RAS* (*N* = 44, 10.3%) genes. The remaining 176 (41.0%) PT had fusion drivers, most commonly involving *RET* (*N* = 73, 17.0%) or other receptor tyrosine kinase (*RTK*) genes (*N* = 64, 14.5%). Among the 356 PT with high-quality WGS data, the low burden of SSVs (mean = 0.29 mutations per Mb) was predominantly comprised of single nucleotide variants (SNVs) (93.3%) and less commonly small insertions (1.8%) or deletions (4.5%). Mutational signature analysis[32,33] identified clock-like signatures as the most commonly occurring single base substitution (SBS) and small indel (ID) mutational signatures (SBS1 = 9.8%, SBS5 = 60.2%; ID1 = 14.2%, ID5 = 39.8%). Over half (*N* = 190, 53.4%) of PT had at least one confirmed structural variant (SV), and 143 (40.3%) had at least one somatic copy number alteration (SCNA), most frequently (*N* = 49, 13.8%) the loss of 22q.

### Modeling of characteristics associated with cLNM occurrence

Among all the patient, clinical, and molecular characteristics we evaluated in sex- and age at PTC-adjusted multivariable models, cLNM occurrence was most strongly associated with PTC driver; there was no measurable effect of cumulative environmental radiation exposure on cLNM occurrence ($P_{trend}$ = 0.32) (Table S3). Specifically, cLNM were notably more common in tumors with fusion than mutation drivers (*N* = 97/156, 55.1% vs. *N* = 76/252, 30.2%; *P* = 5.8 × 10⁻⁶), with differing frequencies among specific drivers ($P_{heterogeneity}$ = 1.6 × 10⁻¹⁹) (Fig. 1 and Table S4). cLNM were most common among tumors with *RET* (*N* = 52/73, 71.2%) or other *RTK* (*N* = 41/64, 64.1%) fusion drivers but less common among tumors with other fusion drivers (*N* = 4/39, 10.3%), all of which occurred in tumors with *BRAF* fusions. Additionally, cLNM were recorded for 73/194 (37.6%) of tumors with *BRAF* mutations but only 3/58 (5.2%) with driver mutations in other genes, most commonly *RAS* (*N* = 2/43, 4.7%). Analyses by the specific fusion partner showed a slightly higher frequency of cLNM occurrence for tumors with *NCOA4-RET* (*N* = 12/15, 80.0%) than *CCDC6-RET* (*N* = 26/40, 65.0%) fusion

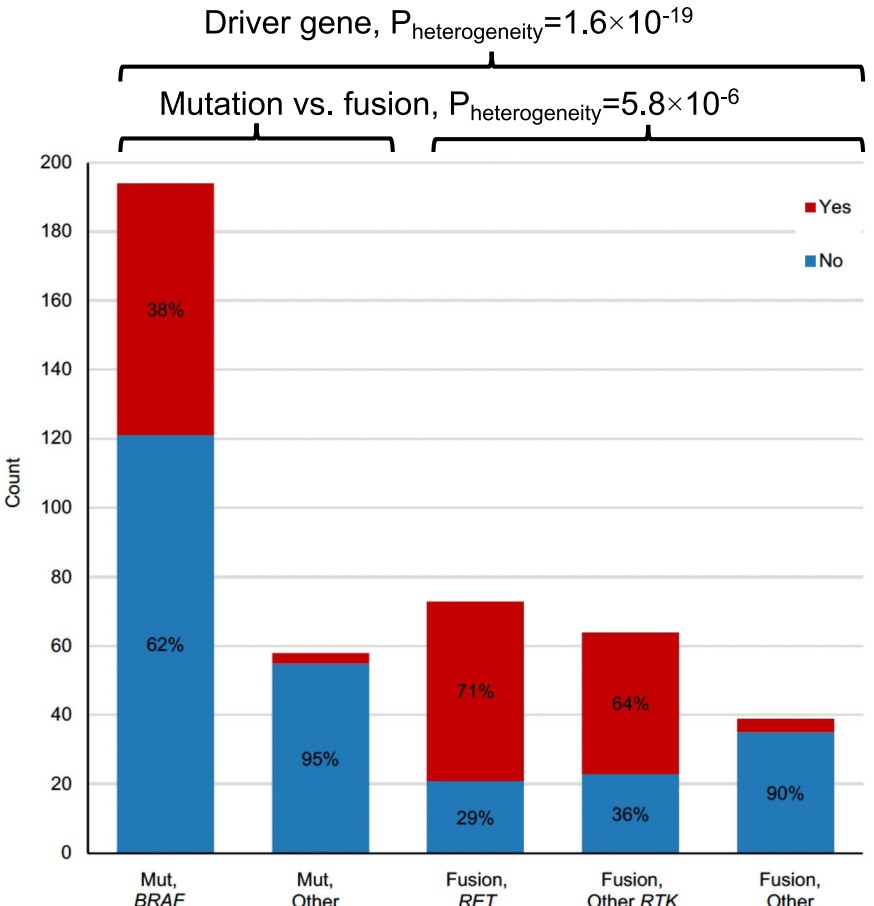

**Fig. 1 | Distribution of cLNM at diagnosis among 428 PTC tumors in our primary study population with a final designated driver, by driver type and gene.** Red = cLNM, Blue = no cLNM. Percentages >20% are shown. $P_{\text{heterogeneity}}$ represents a two-sided $P$ value calculated using likelihood ratio tests, comparing model fit with and without the variable of interest. Source data are provided as follows: Table S3 provides all counts and percentages, including information on the other mutations and fusions.

drivers, and for *NTRK1* ($N = 11/13$, 84.6%) than other *RTK* ($N = 30/51$, 58.8%) fusion drivers, regardless of the fusion partner (Table S5). The excess occurrence of cLNM associated with fusion drivers was more pronounced for N1b than N1a cLNM (N1b: fusion driver $N = 57/156$, 32.4%, mutation driver $N = 32/252$, 12.7%; N1a: fusion driver $N = 40/156$, 22.7%, mutation driver $N = 44/252$, 17.5%), particularly for *NCOA4-RET* (N1b: $N = 9/15$, 60.0%; N1a: $N = 3/15$, 20.0%) (Table S5). Analyses stratified by age at PTC (<30 vs. ≥30 years, reflecting the mean age at PTC diagnosis among exposed individuals) revealed that the frequency of cLNM occurrence was consistent by age in tumors with *RET* fusion drivers (71.2 vs. 71.4%) but declined at older age for tumors with *BRAF* mutation (41.8 vs. 34.0%) and other *RTK* fusion drivers (69.0 vs. 54.5%) (Fig. S2). More detailed breakdowns of cLNM occurrence by driver and age at PTC are provided in Table S6. Although results should be interpreted cautiously due to small numbers of cases in certain subgroups, the patterns observed in our overall study population appeared consistent when we restricted to pediatric cases (<20 years), namely a high proportion of cLNM occurrence in PTC with RET (76.2%) and other RTK (64.3%) fusion drivers and a lower percentage for BRAF mutations (23.5%).

When we re-analyzed patient and clinical predictors of cLNM occurrence separately for tumors with the two most commonly occurring drivers, cLNM occurrence was significantly associated with increasing tumor size for tumors with either *BRAF* mutation ($P_{\text{trend}} = 1.8 \times 10^{-3}$) or *RET* fusion ($P_{\text{trend}} = 9.4 \times 10^{-4}$) drivers, whereas there was a suggestive association with younger age for tumors with *BRAF* mutation drivers only ($P = 0.089$) (Table 1). Specifically, the odds

of cLNM occurrence for tumors >2 versus ≤1.0 cm were nearly three-fold higher for PTCs with a *BRAF* mutation driver (odds ratio [OR] = 2.9, 95% confidence interval [CI] = 1.2–6.8) and over twelve-fold higher for PTCs with a *RET* fusion driver (OR = 12.3, 95%CI = 1.2–112). No dose-response trend was observed between radiation and cLNM in categorical, linear, linear-quadratic, or quadratic models. Additional molecular characteristics were not associated with cLNM occurrence in either driver group ($P > 0.01$) (Table S3).

Previous reports have suggested that *TP53, AKT1, PIK3CA*, and *TERT* promoter mutations, *CDKN2A* deletions, *PTEN* mutations/deletions, *MYC* amplifications, and whole genome duplication are associated with the development of metastases[6,8,17,34–40]. However, we observed only small numbers of each of these alterations, precluding analysis in our dataset (*TP53* mutation, N0 = 2; *AKT1*, N0 = 3; *PIK3CA*, N1 = 1; *TERT* promoter mutation, N1 = 1; *CDKN2A* deletions, N0 = 1; *PTEN* mutations or deletions, N0 = 3; *MYC* amplifications, N0 = 2; whole genome duplication, N0 = 2). We further compared transcriptomic profiles of PT with and without cLNM but did not find any significantly differentially expressed genes (DEGs) in analyses restricted to the two most commonly occurring drivers, *BRAF* mutation and *RET* fusion.

We extended our analysis of the relationship between PTC driver and cLNM occurrence by re-analyzing previously published data from two sources (Fig. S1 and Tables S1, S2): (1) 68 individuals from two previous Chornobyl studies with available $^{131}$I exposure data, restricted to those with known drivers and non-overlapping with our study population[41,42] and (2) 326 individuals from the TCGA analysis, restricting to those with known fusion or mutation drivers, available

**Table 1 | Relationship of patient and pathologic characteristics to the occurrence of cLNM in PTC with *BRAF* mutation or *RET* fusion drivers**

| Characteristic | *BRAF* mutation driver | | | | *RET* fusion driver | | | |
|---|---|---|---|---|---|---|---|---|
| | cLNM | | | | cLNM | | | |
| | No | Yes | OR | (95%CI)* | No | Yes | OR | (95%CI)ᵃ |
| Sex | | | | | | | | |
| Female | 99 | 54 | 1.0 | (referent) | 17 | 39 | 1.0 | (referent) |
| Male | 22 | 19 | 1.4 | (0.6, 2.9) | 4 | 13 | 1.3 | (0.3, 6.1) |
| Age at PTC (years) | | | | | | | | |
| <25 | 25 | 20 | 1.0 | (referent) | 10 | 29 | 1.0 | (referent) |
| 25–29 | 28 | 18 | 0.6 | (0.2, 1.6) | 7 | 13 | 0.6 | (0.2, 2.6) |
| ≥30 | 68 | 35 | 0.4 | (0.2, 1.0) | 4 | 10 | 1.2 | (0.2, 7.2) |
| $P_{trend}$ | | | | 0.089 | | | | 0.74 |
| Radiation dose (mGy)ᵇ | | | | | | | | |
| 0 | 24 | 11 | 1.0 | (referent) | 5 | 11 | 1.0 | (referent) |
| 1–99 | 66 | 40 | 2.2 | (0.8, 6.1) | 8 | 15 | 0.9 | (0.1, 6.2) |
| 100–199 | 18 | 17 | **3.4** | **(1.1, 10.8)** | 4 | 10 | 2.0 | (0.3, 13.8) |
| ≥200 | 13 | 5 | 1.4 | (0.3, 5.6) | 4 | 16 | 2.1 | (0.4, 11.7) |
| $P_{trend}$ | | | | 0.92 | | | | 0.13 |
| Multifocal lesion | | | | | | | | |
| No | 100 | 54 | 1.0 | (referent) | 18 | 48 | 1.0 | (referent) |
| Yes | 21 | 19 | 1.9 | (0.9, 4.1) | 3 | 4 | 0.2 | (0.02, 1.8) |
| Primary lesion size (cm) | | | | | | | | |
| ≤1.0 | 38 | 16 | 1.0 | (referent) | 3 | 5 | 1.0 | (referent) |
| >1.0–2.0 | 61 | 32 | 1.2 | (0.6, 2.6) | 15 | 19 | 1.0 | (0.2, 5.3) |
| >2.0 | 22 | 25 | **2.9** | **(1.2, 6.8)** | 3 | 28 | **12.3** | **(1.3, 112)** |
| $P_{trend}$ | | | | $1.8 \times 10^{-3}$ | | | | $9.4 \times 10^{-4}$ |

*cLNM* cervical lymph node metastasis, *CI* confidence interval, *OR* odds ratio, *PTC* papillary thyroid carcinoma.

Bolded font represents $P < 0.05$. Source data are provided in Data S2.

ᵃMultivariable logistic regression model included all patient and pathologic characteristics in the table. $P_{trend}$ represents a two-sided *P* value calculated using likelihood ratio tests, comparing model fit with and without the variable of interest.

ᵇNo significant improvement in model fit was observed when considering radiation dose in a linear-quadratic (*BRAF* mutation: $P = 0.14$; *RET* fusion: $P = 0.82$) or quadratic (*BRAF* mutation: $P = 0.58$; *RET* fusion: $P = 0.14$) model. No radiation dose-response trend among the categories was observed when we excluded unexposed individuals from the model (reference = 1–99 mGy; *BRAF* mutation: $OR_{100–199mGy} = 1.7$, 95% CI = 0.7–4.0, $OR_{≥200mGy} = 0.6$, 95% CI = 0.2–2.0; *RET* fusion: $OR_{100–199mGy} = 2.6$, 95% CI = 0.4–17.0, $OR_{≥200mGy} = 2.2$, 95% CI = 0.4–12.9).

pathology data, and without known radiation exposure[17]. Although the sample size from the previous Chornobyl studies was limited, the patterns of cLNM occurrence by driver in those studies, as well as TCGA, generally were consistent with our original study population, though only the differences in cLNM occurrence by the specific driver in TCGA were statistically significant (Fig. S3 and Table S4). Namely, cLNM were more frequent in tumors with fusion than mutation drivers (Chornobyl: 47.9 vs. 30.0%, $P = 0.81$; TCGA: 50.9 vs. 44.9%, $P = 0.44$), and specifically in tumors with *RET* fusion than *BRAF* mutation drivers (Chornobyl: 55.2 vs. 37.5%, among all specific drivers: $P_{heterogeneity} = 0.39$; TCGA: 78.3 vs. 50.2%, among all specific drivers: $P_{heterogeneity} = 1.4 \times 10^{-5}$). Similarly, pooled analyses of patient and clinical characteristics demonstrated a consistently increased occurrence of cLNM associated with larger tumor size and more advanced stage for both *BRAF* mutation and *RET* fusion-driven tumors among all three studies (Tables S7, S8). In contrast, associations with male sex and younger age in individuals with *BRAF* mutation-driven tumors were inconsistent.

## Molecular profiles of cLNM samples

We comprehensively characterized the molecular profiles of 47 fresh-frozen, pre-treatment cLNM samples with high-quality WGS (mean

sequencing depth = 89X), mRNA-seq ($N = 46$; average read count = 135 million per sample), miRNA-seq ($N = 43$), DNA methylation profiling (Illumina Infinium MethylationEPIC array; $N = 45$), and relative telomere length ($N = 43$) (Supplementary Data 3). Comparable data from paired PT samples have been published previously[18]. Figure S4 and Table S9 detail the availability of samples by type and platform. Although the available cLNM samples had a higher proportion of fusion-driven tumors (Table S9), the distributions with respect to patient age, sex, and radiation dose were otherwise comparable (Table S10). For each data type, we confirmed the lack of additional lymphocyte infiltration in our cLNM compared to PT samples, as detailed below.

## cLNM driver identification

We identified drivers using WGS and mRNA-seq for mutation and fusion detection[18], and all cLNM samples had the same final designated driver as the paired PT (Fig. 2 and Table S9). A total of 2/47 (4.3%) cLNM samples, both with *RET* fusion drivers, had *MYC* amplifications that were not present in their paired PT samples; conversely, the *TERT* promoter mutation observed in the single PT sample from our previous study was not present in its paired cLNM sample. No other new putative driver mutation private to the cLNM sample was identified.

## cLNM genomic characteristics

We comprehensively compared the genomic characteristics of 45 cLNM and 41 paired PT samples with high tumor purity and high-quality WGS data (Fig. S4). To increase statistical power, we further selected another 85 high-purity and high-quality PT samples from other individuals from our original study of 440 individuals[18], matched on PTC driver (Supplementary Data 1 and Table S9).

In unpaired analyses ($N = 45$ cLNM, $N = 126$ PT), cLNM and PT had comparably low burdens of SSVs (mean = 0.27 mutations per Mb in both sample types), with similar distributions of SNVs (cLNM = 92.6%, PT = 93.2%), small insertions (cLNM = 2.0%, PT = 1.7%) and deletions (cLNM = 5.1%, PT = 4.7%), and multiple base substitutions (cLNM = 0.4%, PT = 0.4%) (Figs. 2, 3A–C and Table S11). No additional SBS or ID mutational signatures were identified in the cLNM samples, and the distributions of specific SBS and ID signatures were similar between cLNM and PT. We also observed comparable numbers of samples with at least one SV (cLNM: $N = 35/45$, 77.8%; PT: $N = 104/126$, 82.5%) or SCNA (cLNM: $N = 17/45$, 37.8%; PT: $N = 51/126$, 40.5%), with 22q loss in a small minority (cLNM: M = 2/45, 4.4%; PT: N = 11/126, 8.7%). None of the cLNM samples exhibited whole genome duplication. The mean fraction of the genome altered was similar between cLNM (mean = 0.44 ± 1.35%) and PT (mean = 0.30 ± 0.80%). In multivariable regression models adjusted for age and sex, the distribution of the measured genomic characteristics was not statistically different between cLNM and PT samples, except the relative telomere length was suggestively shorter in cLNM ($P = 0.014$) (Table S12). Results generally were comparable when we further adjusted our initial models for radiation dose and driver, and when we restricted analyses to the paired cLNM-PT samples ($N = 41$) from the same individuals (Fig. S5 and Table S12), except the suggestively shorter relative telomere length in cLNM was no longer evident in the paired analysis ($P = 0.24$).

Within the paired cLNM-PT samples, we characterized SNVs by whether they were shared versus private to the cLNM or PT samples as well as by clonality (clonal = cancer cell fraction ≥ 0.6, subclonal = cancer cell fraction <0.6). SNVs were approximately evenly distributed among shared (mean, range: 34.3%, 7.8–75.9%), private cLNM (30.7%, 5.0–65.5%), and private PT (35.0%, 8.3–63.0%) mutations (Fig. S6 and Table S13). As expected, the shared SNVs were more frequently clonal (mean, clonal = 29.4% vs. subclonal = 3.2%), whereas the private cLNM and private PT SNVs were more frequently subclonal (private cLNM: clonal = 6.0% vs. subclonal = 25.4%; private PT: clonal = 5.6% vs. subclonal = 30.4%).

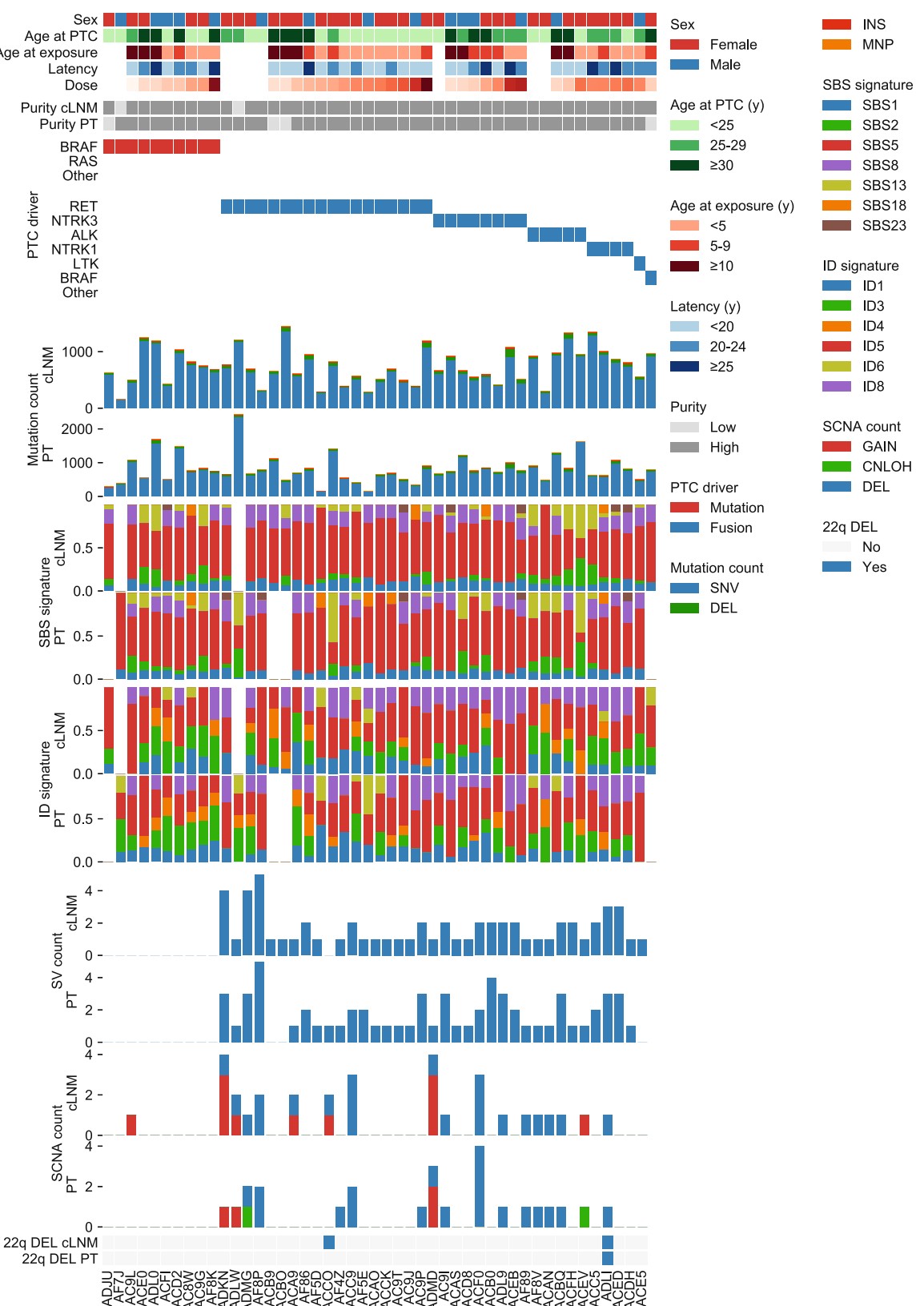

**Fig. 2 | Comparison of genomic characteristics between 47 cLNM and paired PT samples.** Data include patient and sample characteristics, PTC driver, mutation, structural variant, and somatic copy number alteration counts. Source data are provided as follows: Table S9 provides all counts.

To investigate patterns of metastasis seeding, we specifically focused on clonal private and subclonal shared SNVs (Figs. S6, S7)[43,44]. The presence of subclonal shared SNVs (*N* ≥ 10 SNVs) in 22 paired cLNM-PT samples is indicative of polyclonal seeding. For the remaining 19 samples, the more dominant presence of private clonal mutations in the cLNM and/or PT is evidence of sampling bias, which obfuscates the metastasis polyclonal versus monoclonal seeding pattern and likely occurred because we sampled and sequenced only a

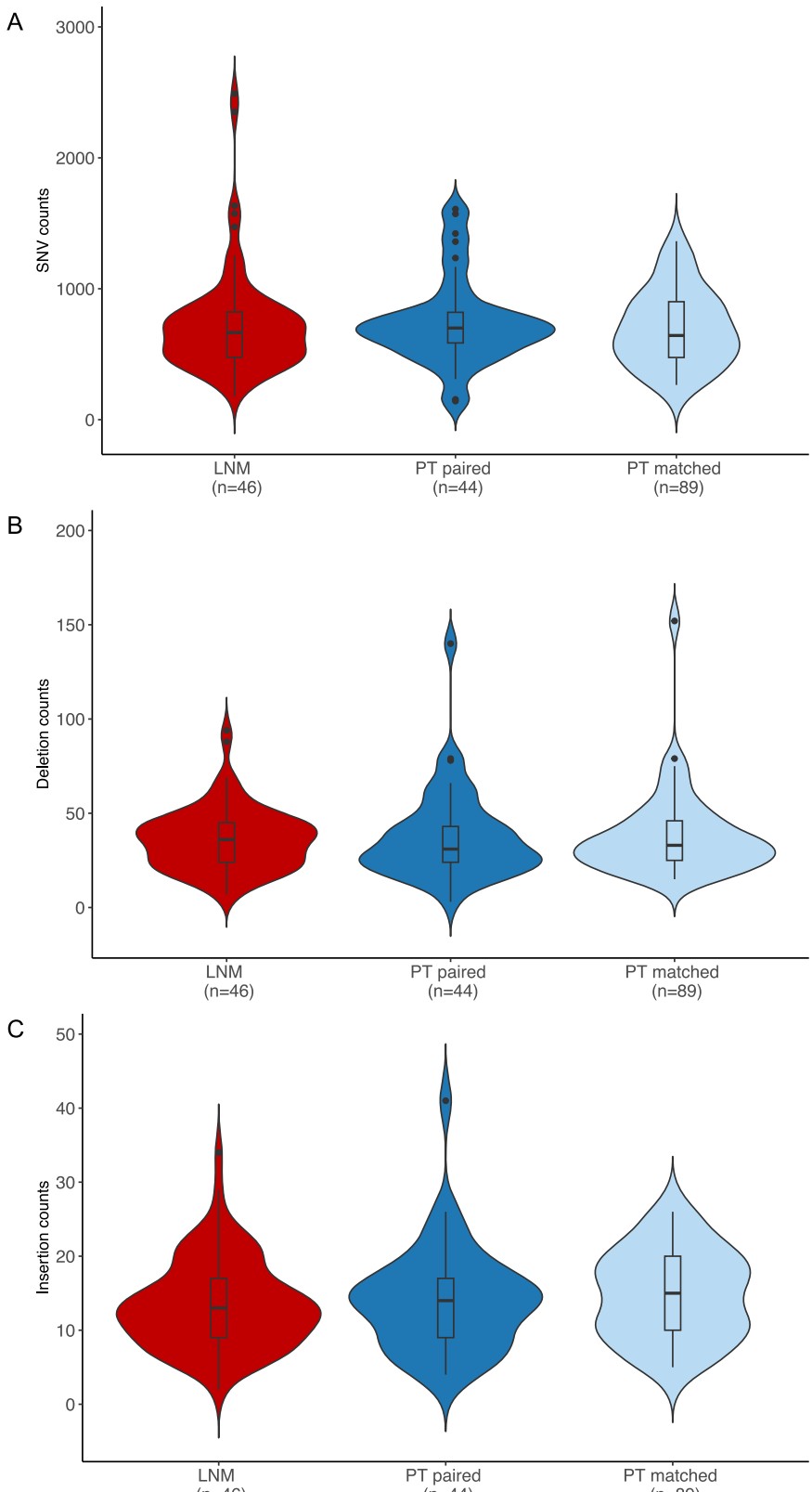

**Fig. 3 | Burden of simple somatic variants (SSVs) by sample type.** Distribution of SNVs (**A**), small deletions (**B**), and small insertions (**C**) in cLNM and PT samples. Red = cLNM, Blue = paired primary tumors, Light blue = matched primary tumors. All box plots include the center line at the median, the box denotes the interquartile range (IQR), whiskers denote the rest of the data distribution, and outliers are denoted by points greater than ±1.5 × IQR. Note that the Y-axis scales differ among panels. Source data are provided as follows: Table S9 provides counts. Figure S5 provides scatter plots of the distributions in paired cLNM-PT samples.

single area within both the cLNM and PT. The similarity of the tumor purity between the cLNM and PT samples supports a lack of additional lymphocyte infiltration in the cLNM samples.

## cLNM transcriptomic characteristics

We compared the transcriptomic characteristics of cLNM and PT, restricting analyses to 46 cLNM samples with mRNA-seq data, paired PT samples ($N$ = 44), as well as 89 additional PT samples from our original study of 440 individuals[18], matched on PTC driver (Supplementary Data 1, Fig. S4, and Table S9). We observed strong evidence in three genes for substantive differential expression: overexpression of *HOXC10* (log(2) fold change [log2FC] = 3.4, false discovery rate [FDR]-adjusted $P$ value ($P_{adj}$) = 6.4 × 10$^{-23}$) and *HOTAIR* (log2FC = 2.9, $P_{adj}$ = 2.9 × 10$^{-17}$) and underexpression of *BRINP3* (log2FC = −3.6, $P_{adj}$ = 1.3 × 10$^{-17}$) in cLNM compared with PT samples (Fig. 4A and Supplementary Data 4). Both *HOXC10* and *HOTAIR*, as well as seven other genes with $P_{adj}$ < 1.0 × 10$^{-4}$ and log2FC >1.0, are part of the *HOXC* locus on 12q13.13. Notably, SCNAs are unlikely to be the mechanism of the *HOXC* locus dysregulation as no SCNAs occurred across the *HOXC* locus in either the cLNM or paired PT samples. Correlation analyses in cLNM samples for the expression levels of these genes demonstrated two main blocks of correlated genes, both of which included *HOXC10* (Fig. 4B). After adjusting for *HOXC10* expression, none of the remaining genes in the *HOXC* locus were significantly differentially expressed between cLNM and PT samples (Supplementary Data 4), supporting further investigation of regulatory elements in or near the *HOXC10* region of the locus. Analyses of the expression levels for our top three genes revealed overexpression in *HOXC10* and *HOTAIR* only in cLNM but not PT or non-tumor thyroid samples (Fig. 5A, B), whereas *BRINP3* expression was highest in non-tumors thyroid samples, moderate in PT samples, and very low in cLNM (Fig. 5C). Overall, our differential expression results were similar when we further adjusted our initial models for radiation dose and driver, and when we restricted analyses to the cLNM and paired PT samples from the same individuals (Supplementary Data 4). Further analyses of gene expression using the Molecular Signatures Database (MSigDB) Hallmark gene sets[45,46] did not yield any statistically significant differences between cLNM and PT (Supplementary Data 5).

In parallel analyses of miRNA, we identified four miRNAs that were substantially differentially expressed: overexpression of miR-196a2 (log2FC = 3.9, $P_{adj}$ = 1.3 × 10$^{-25}$) and miR-615 (log2FC = 2.3, $P_{adj}$ = 3.2 × 10$^{-18}$) and underexpression of miR-137 (log2FC = −2.5, $P_{adj}$ = 7.7 × 10$^{-13}$) and miR-141 (log2FC = −1.0, $P_{adj}$ = 1.6 × 10$^{-12}$) in cLNM compared with PT samples (Figs. 6, 7A−D and Supplementary Data 6). miR-196a2 and miR-615 have correlated expression ($r$ = 0.67) and are located in the *HOXC* locus near *HOXC10* and *HOXC5*, respectively. Further adjustment of models for radiation dose and driver, as well as restriction of analyses to the cLNM and paired PT samples from the same individuals, generally yielded similar results (Supplementary Data 6). Exploratory analyses of the top differential expression results for mRNA (*HOXC10*) and miRNA (miR-196a2) revealed consistent findings when we restricted to pediatric cases (<20 years) (Fig. S9).

Several lines of evidence provide support that our findings were not due to additional lymphocyte contamination in the cLNM compared with PT samples. Unsupervised mRNA-seq clustering analyses showed extensive overlap between cLNM and PTC samples, suggesting lymphocyte levels did not substantially differ between the sample groups (Fig. S8). Both Genotype-Tissue Expression (GTEx) project bulk sequencing data and a previously published analysis of single-cell mRNA-seq across tissue types show that *HOXC10*, *HOTAIR*, and *BRINP3* are not highly expressed in lymphocytes (Fig. S10)[47,48]. Finally, the application of CIBERSORTx to identify cell types similarly suggested comparable immune cell distributions in cLNM and PT samples (Fig. S11)[49,50].

## cLNM epigenomic characteristics

We compared the epigenomic characteristics of cLNM and PT based on 43 cLNM and 36 high-purity paired PT samples with DNA methylation data, as well as an additional 86 high-purity PT samples, matched on PTC driver (Supplementary Data 1, Fig. S3, and Table S9). Comparing differential DNA methylation between cLNM and PT, a total of 68 of the 783,071 high-quality probes had $P_{adj}$ < 0.05, though none had $P_{adj}$ < 1.0 × 10$^{-4}$, indicating no differences in epigenomic characteristics at this higher significance level, even for probes in the region of *HOXC* at 12q13.13 (Supplementary Data 7).

The lack of additional lymphocyte contamination in the cLNM compared with PT samples was further supported by two epigenomic analyses. Investigation of the cell type composition showed comparable overlap of the characteristic immune cell epigenomic profiles of matched cLNM and PT samples (Fig. S12), and the 68 probes with $P_{adj}$ < 0.05 showed no enrichment for active immune regions (Fig. S13)[51,52].

## Discussion

In this study, we investigated the association between cLNM occurrence and a range of patient, clinical, epidemiologic, and molecular characteristics using genomic landscape data. The large ($N$ = 440) number of patients ascertained with centralized pathology review and consistent surgical management from a single large tertiary center[18–20], detailed clinical information, and quantitative radiation exposure data represent an opportunity to improve understanding of cLNM in PTC. In patients predominantly diagnosed during young adulthood, we demonstrate that the PTC driver is the dominant factor associated with cLNM, with the highest frequency of cLNM in tumors with *RET* or *NTRK1* fusion drivers. Based on multivariable models that included genomic landscape data, our findings suggest that prior reports of strong associations of age at PTC and prior radiation exposure with cLNM occurrence were likely influenced by the relationship of these variables with the PTC driver. Our comprehensive interrogation of cLNM samples revealed the lack of a second, novel driver and similar mutational spectra compared with PT samples. However, transcriptomic changes centered on the *HOXC* cluster on chromosome 12q13.13, as supported by both mRNA-seq and miRNA-seq results, provide direction for future research on the biological underpinnings of PTC cLNM.

Using multivariable modeling, we found that the PTC driver was the strongest predictor of cLNM occurrence among all the patient, clinical, and molecular characteristics. The high frequency of cLNM among PTC with *RET* (71.2%) or other *RTK* (64.1%) fusion drivers—especially *NTRK1* (84.6%)—persisted regardless of age at diagnosis, in contrast to the lower frequency (37.6%) of cLNM among PTC with *BRAF* mutation drivers, which declined even further with increasing age. Future research is needed to understand the higher metastatic potential of RTKs, which influence multiple different signaling pathways, in contrast to the lower metastatic potential of other drivers, such as *BRAF* fusion, *BRAF* mutation, and *RAS* mutation, which primarily regulate the MAPK pathway specifically. Overall, our results provide a valuable bridge between previous pediatric PTC studies[14–16] and the predominantly older adult PTC in the TCGA study[17]. Our findings regarding PTC driver and cLNM frequency were consistent with those from pediatric PTC studies, although our reanalysis of the previous Chornobyl studies with a younger mean age at PTC, found a non-significant increased frequency of cLNM in fusion-driven tumors, perhaps due in part to smaller sample size. Nevertheless, our results suggest that patterns of cLNM among young adults are more similar to those of pediatric rather than older adult patients. The slightly higher frequency of cLNM, particularly N1b, that we observed for tumors with *NCOA4-RET* than *CCDC6-RET* fusion drivers, albeit based on small numbers, also is consistent with previous reports[53,54] of increased aggressiveness of tumors with *RET/PTC3* rearrangements. After model

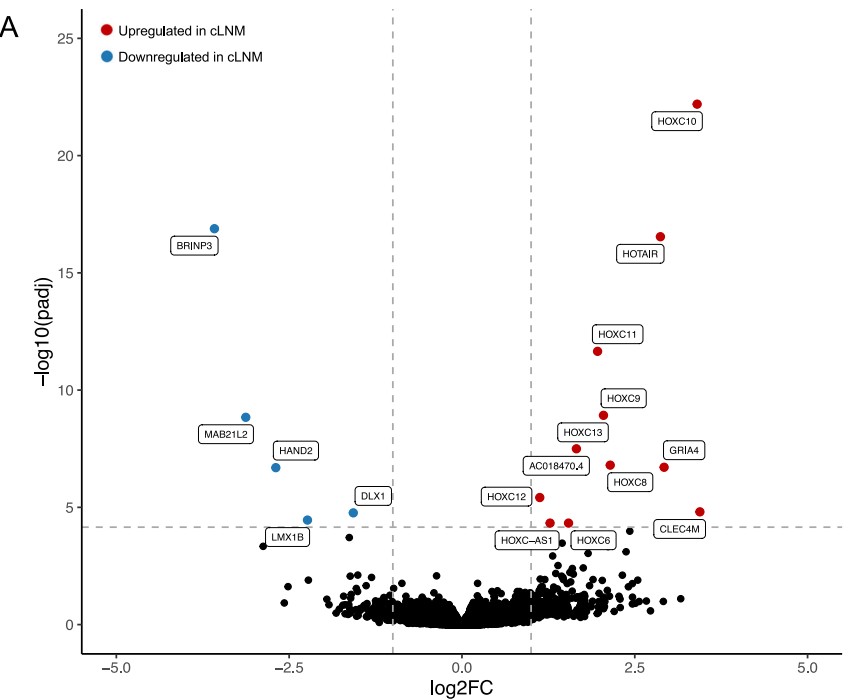

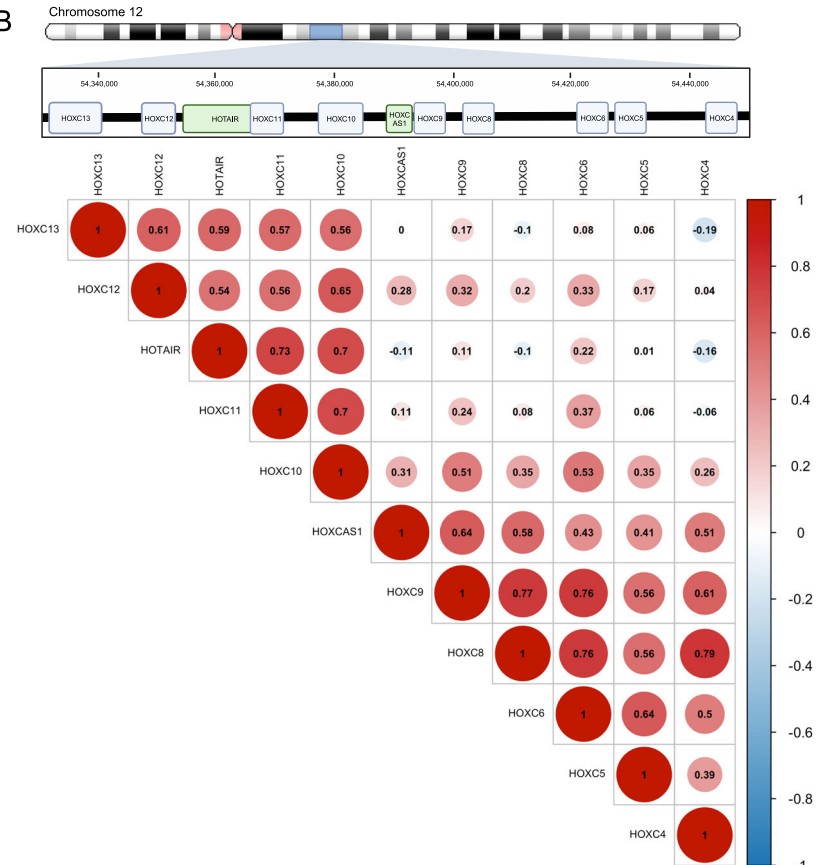

**Fig. 4 | Results of transcriptomic analyses.** Differentially expressed genes between cLNM and PT samples (**A**) and correlation among gene expression levels within the *HOXC* locus (created with BioRender.com) (**B**). Two-sided *P* values were calculated using simple linear regression models on the normalized read counts to determine whether tissue status (cLNM vs. PT) was associated with differential gene and mRNA expression, adjusted for sample batch, sex, and age at PTC. Adjusted *p* values were calculated using the standard Benjamini–Hochberg false discovery rate (FDR) method. Source data are provided as follows: Supplementary Data 4 provides the full results.

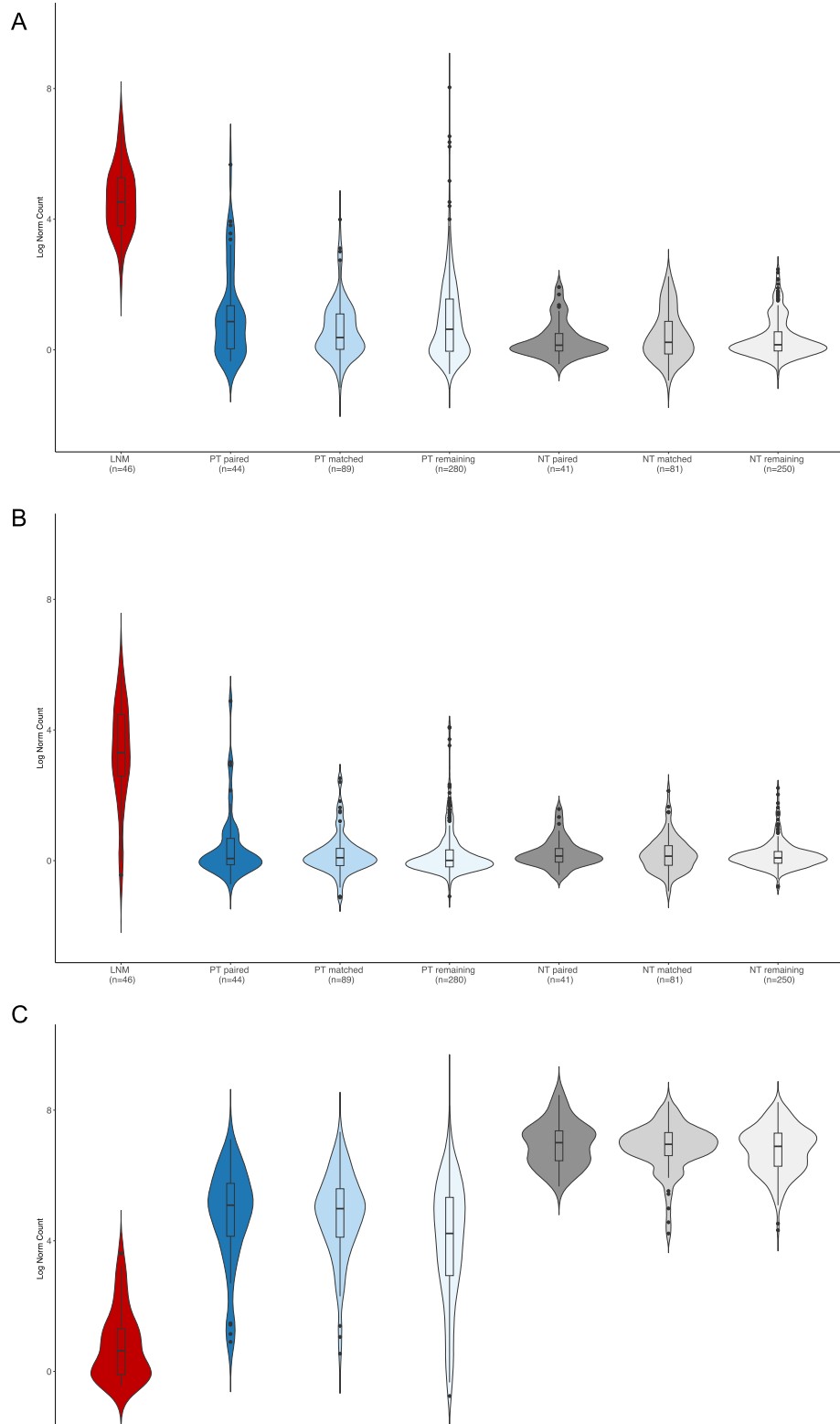

**Fig. 5 | Top differentially expressed genes.** Expression levels of *HOXC10* (**A**), *HOTAIR* (**B**), and *BRINP3* (**C**) in each sample type. Red = cLNM, Shades of blue = primary tumors, Shades of gray = non-tumor thyroid tissue. All box plots include the center line at the median, the box denotes the interquartile range (IQR), whiskers denote the rest of the data distribution, and outliers are denoted by points greater than ±1.5 × IQR. Source data are provided as follows: Supplementary Data 4 provides the full results.

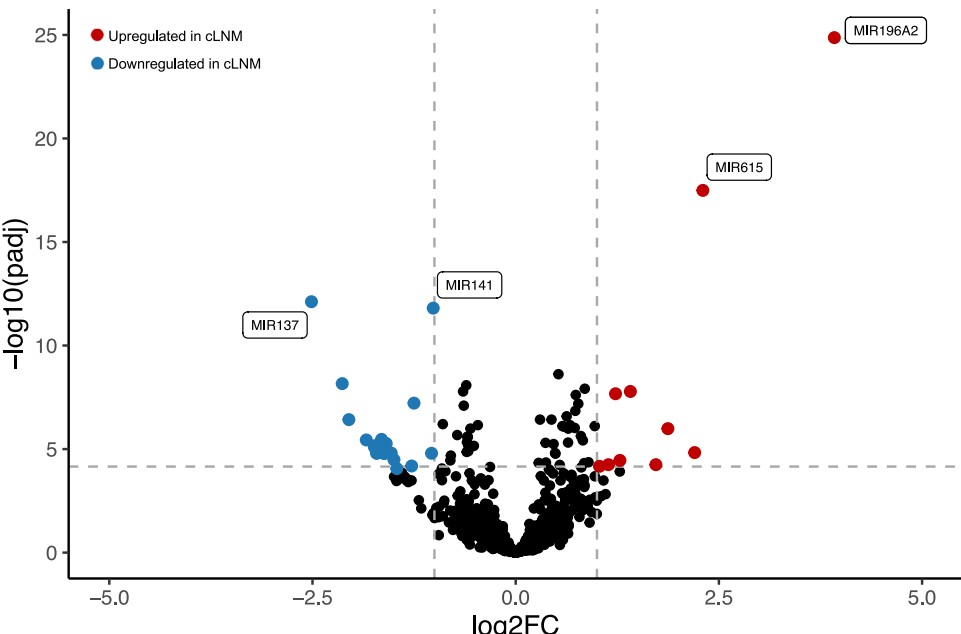

**Fig. 6 | Results of miRNA analyses.** Differentially expressed miRNAs between cLNM and PT samples. Two-sided *P* values were calculated using simple linear regression models on the normalized read counts to determine whether tissue status (cLNM vs. PT) was associated with differential gene and miRNA expression, adjusted for sample batch, sex, and age at PTC. Adjusted *p* values were calculated using the standard Benjamini–Hochberg false discovery rate (FDR) method. Source data are provided as follows: Supplementary Data 6 provides the full results.

adjustment for the PTC driver, only larger tumor size remained consistently associated with the development of cLNM. Notably, prior studies that lacked comprehensive driver data and/or multivariable modeling have reported that younger age at diagnosis, male sex, and exposure to ionizing radiation were associated with increased cLNM occurrence[4,5,22–27]. However, the increased frequency of fusion drivers among young individuals and those exposed to higher doses of radiation suggests that prior reports associating these characteristics with cLNM occurrence were likely influenced by the relationship of these variables with the PTC driver. Together, our data support the importance of including molecular markers, specifically the tumor driver, in the clinical management of PTC.

Molecular profiling of 47 cLNM samples yielded three main observations: (1) 100% concordance of the driver alterations, (2) no novel drivers, and (3) highly concordant mutational spectra compared with the matched PT. The lack of new driver mutations in the cLNM supports the central role of the primary PTC driver in the development of local metastases, while the lack of evidence for late-hit mutations in cLNM in our study further supports the similarity of young adult PTC with pediatric PTC, as late-hit mutations have predominantly been reported in distant metastases among older patients[6,17].

In transcriptomic analysis of PT and cLNM, we identified a number of differentially expressed genes and miRNAs. Most notably, we observed overexpression in cLNM of *HOXC10, HOTAIR*, and seven other genes in the 12q13.13 *HOXC* locus, as well as two miRNAs in the same locus (miR-196a2 and miR-615). Conditional mRNA analyses of the locus demonstrated that *HOXC10* retained the strongest effect and thus likely harbors the regulatory regions most promising for further investigation. *HOXC* includes a set of highly conserved genes that are part of the homeobox family of transcription factors and have been implicated broadly in carcinogenesis, although the specific mechanisms are not well understood[55,56]. A plausible role for *HOXC* genes, including *HOXC10*[57] and particularly *HOTAIR*[58,59], in PTC metastasis is further supported by reports of increased *HOXC10* expression in clinically aggressive thyroid cancer[60]; association of increased *HOTAIR*

expression with tumor size, pathologic stage, and cLNM in thyroid cancer[61–64]; and evidence that *HOTAIR* promotes migration and invasion of thyroid cancer cell lines[65,66]. *HOTAIR* also is increasingly recognized as a critical contributor in the metastatic process for a number of cancers more broadly, specifically due to its role as a regulator of epithelial cell plasticity and the epithelial-to-mesenchymal transition (EMT)[67], although our GSVA analyses did not identify major differences in gene expression in the EMT or other hallmark pathways overall. Much less is known about the function of *BRINP3*, a developmental gene whose reduced expression in cLNM was the second strongest DEG in our analysis, though it has been implicated as a cell cycle regulator and associated with cellular proliferation and migration in osteosarcoma[68]. Intriguingly, two other genes (*DLX1, LMX1B*) that also had lower expression in the cLNM are homeobox genes[69], suggesting the potential importance of dysregulation of developmental and differentiation processes in PTC cLNM. These associations are further supported by our observation of reduced expression of miR-137 and miR-141, both of which are tumor suppressor miRNAs that reportedly play a role in cancer occurrence and progression[70,71]. miR-141, in particular, is part of the miR-200 family, which has been shown to target and inhibit the ZEB1 and ZEB2 EMT transcription factors and is well described in cancer metastasis[71,72]; miR-141 also specifically has been shown to be downregulated in thyroid cancer, with correlations between expression and cellular proliferation, apoptosis, and migration[73]. Overall, our findings point to promising candidates for future investigations aimed toward elucidating the biological underpinnings of PTC cLNM.

Several points should be considered in the interpretation of our results. Analyses based on data generated across several sequencing platforms did not support additional lymphocyte infiltration in cLNM compared to PT samples. The paucity of distant metastases was notable in our study population, though not all patients were systematically scanned for distant events. Future efforts should comprehensively evaluate clinical-histopathological characteristics and molecular-genetic alterations in relation to updated PTC pathologic classifications[74]. Finally, caution is warranted when comparing

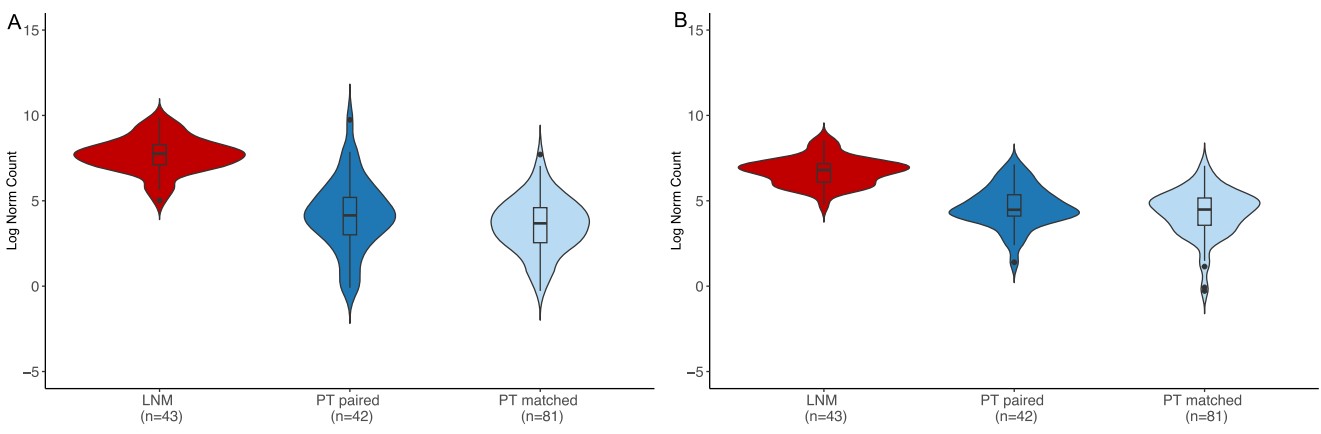

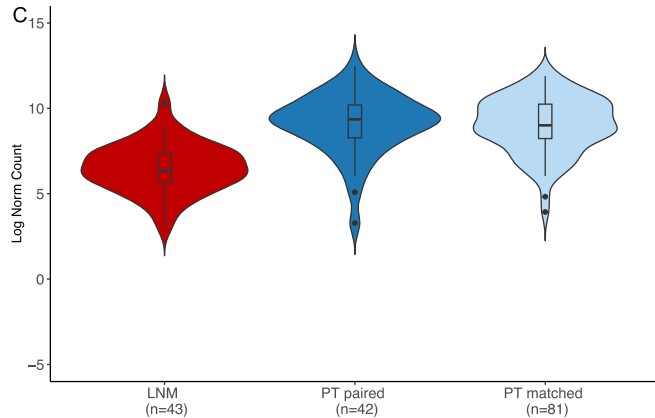

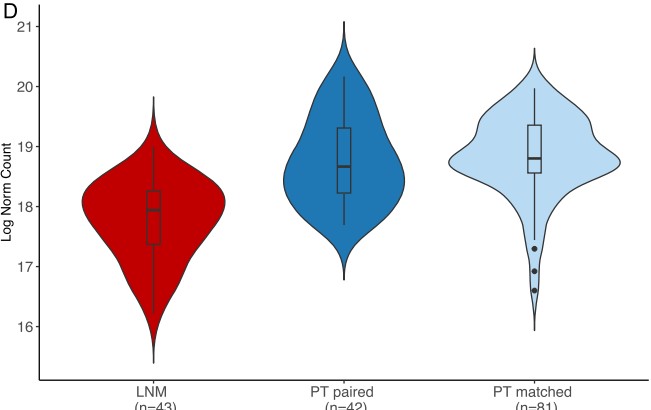

**Fig. 7 | Top differentially expressed miRNAs.** Expression levels of miR-196a2 (**A**), miR-615 (**B**), miR-137 (**C**), and miR-141 (**D**) in each sample type. Red = cLNM, blue = paired primary tumors, light blue = matched primary tumors. All box plots include the center line at the median, the box denotes the interquartile range (IQR), whiskers denote the rest of the data distribution, and outliers are denoted by points greater than ±1.5 × IQR. Source data are provided as follows: Supplementary Data 6 provides the full results.

mutational spectra because we sampled only a single area within each of the cLNM and PT[75–77].

In conclusion, based on comprehensive genomic landscape analyses combined with detailed clinical and patient data, we demonstrate that the PTC driver is directly related to the occurrence of cLNM. Our findings underscore the importance of molecular subtyping in the clinical management of PTC across the age spectrum and inform current approaches to patient risk stratification[78,79]. To better understand the biological underpinnings of metastatic PTC, further investigation of the *HOXC* locus and other differentially expressed genes and miR-NAs is warranted.

## Methods

### Study population

Written informed consent for donation and research use was obtained from all participants or their legal representatives through the CTB. The study was approved by Institutional Review Boards at the patient treatment and tissue collection center (Institute of Endocrinology and Metabolism in Kyiv, Ukraine), the CTB coordination center (Imperial College of London, UK), and the United States National Cancer Institute.

The primary study population included 440 individuals who were diagnosed with pathologically-confirmed papillary thyroid cancer (PTC) during young adulthood in Ukraine following the April 1986 Chornobyl nuclear power plant accident and whose pre-treatment,

fresh frozen tumor sample was collected by the Chornobyl Tissue Bank (CTB)[18–20]. The study population included a total of 359 individuals who were in utero or ≤18 years of age on April 26, 1986 and lived in one of the regions of Ukraine most highly contaminated with [131]I-containing fallout from the accident, and 81 individuals who were born >9 months after the accident and were therefore considered [131]I-unexposed. Quantitative radiation dose estimation was conducted by an international team of dosimetry experts[18,28–30]. Doses were estimated for 49 individuals based on individual thyroid radioactivity measurements taken in May-June 1986, personal interviews regarding residential history and intake of milk and green leafy vegetables, and results of radioecological modeling; for four individuals based on individual thyroid radioactivity measurements but not on personal interview; for 297 individuals based on measurements on different individuals who lived in the same residential area; and for nine individuals who were in utero at the time of the accident based on the mother's exposure.

All patients were evaluated and treated at a single central tertiary medical center, with diagnoses of first primary PTC histopathologically confirmed based on the review of tumor tissue by an international panel of experts through the CTB. Pathologic stage was assigned uniformly by a single expert (T.I.B.) according to the 7th edition of the TNM Classification of Malignant Tumors[31,80]. Individuals with unknown values for nodal (Nx) or metastatic (Mm) were combined with N0 and M0, respectively, for the purposes of these analyses. TNM distributions are provided in Fig. S1 and Tables S1, S2. Among the 440

individuals in our primary study population, 179 (40.7%) had metastases at the time of PTC diagnosis, including 164 (37.3%) with cervical lymph node metastases (cLNM) only (N1M0), 14 (3.2%) with both cLNM and distant metastases (N1M1), and 1 (0.2%) with distant metastases only (N0M1; this individual was excluded from further analyses of cLNM occurrence).

For reference, when the PTCs were staged according to the TNM 8th edition[81], tumors with minimal extrathyroidal extension in the fat and connective tissue that were classified as T3 in the 7th edition were re-classified according to tumor size. Specifically, 71 PTCs shifted from T3 to T1 and 34 from T3 to T2. However, we retained our primary analysis using the TNM 7th edition classification because we analyze tumor size separately and to enable comparison of our results with two previous studies in the literature[17,41,42], both of which were classified using the 7th edition.

## Laboratory methods
We identified a total of 49 fresh-frozen cLNM samples for potential inclusion in the current study. All laboratory methods for sample handling, nucleic acid extraction, library preparation, and sequencing followed those previously described for the primary tumor samples[18], as outlined below.

**Sample handling and nucleic acid extraction.** The fresh-frozen cLNM specimens were obtained from the CTB and processed by Nationwide Children's Hospital (NCH) Biospecimen Core Resource (BCR) (Columbus, Ohio). Dual DNA and RNA extraction was performed utilizing the AllPrep DNA/RNA Mini Kit (Qiagen) for DNA and mirVana miRNA Isolation Kit (Applied Biosystems) for RNA according to manufacturers' instructions. Following extraction, purified nucleic acids were evaluated for quality and quantity.

**Whole genome sequencing (WGS) and library construction.** WGS library preparation and sequencing was performed at the Broad Institute[18]. Libraries were constructed and sequenced on the Illumina HiSeqX with the use of 151 base pair (bp) paired-end reads then processed by the Picard data-processing pipeline. Sequencing data were aligned to the human reference genome (hg19; gs://firecloud-tcga-open-access/tutorial/reference/Homo_sapiens_assembly19.fasta) using BWA-MEM[82]. Each WGS sample was assessed for all quality control processes, including coverage, library complexity, fingerprinting across tumor and normal samples, sequencing error rates, fragment length, chimeric fragment rate, and DNA oxidative damage using a suite of Picard tools (Picard CollectWGSmetrics, CollectSequencingArtifactMetrics for 8-oxoG damage, and CrosscheckFingerprints).

**RNA library preparation and sequencing.** mRNA-seq and miRNA-seq were performed at the Cancer Genomics Research Laboratory (CGR) of the National Cancer Institute[18]. Briefly, libraries were prepared using the Kapa RNA HyperPrep Kit with RiboErase (Kapa Biosystems) and sequenced on the Illumina HiSeq 2500. Trimmed reads were aligned to the GRCh38 human reference genome (Illumina iGenomes NCBI GRCh38) using STAR/2.5.4a. Code for mRNA-seq quality control and alignment is publicly available[83]. For miRNA-seq, ribosomal RNA depletion from 500 ng purified RNA was performed using the Illumina Ribo-Zero Gold rRNA Removal Kit (Illumina), libraries were prepared using the NEBNext Multiplex Small RNA Library Prep Set for Illumina kit. Single-end miRNA reads were processed according to the ENCODE miRNA-seq pipeline (https://www.encodeproject.org/microrna/microrna-seq/) and also mapped to the human hairpin sequences in reference microRNA database "miRbase" version 22.1 (https://www.mirbase.org/) using a custom pipeline[84]. FASTQC/0.11.5 was used for the quality control analysis of post-trimmed RNA-seq and miRNA-seq reads. Samples that received warning or fail messages on the analysis of Mean Quality Scores, per sequence quality scores, per base N content, or adapter content were filtered. STAR alignment scores were used for alignment quality control analysis. Samples over 60% of any type of unmapped reads were filtered.

**Relative telomere length measurement.** We measured relative telomere length using a qPCR assay[18]. In brief, we measured the ratio of telomere (T) signals specific to the telomere hexamer repeat sequence TTAGGG to autosomal single copy gene (S) signals, and then standardized this ratio using internal control DNA samples to yield relative standardized T/S ratios proportional to average telomere length.

**MethylationEPIC array.** High-throughput genome-wide methylation analysis was performed on bisulfite-treated DNA (EZ-96 DNA Methylation MagPrep Kit (Zymo Research)) at the Cancer Genomics Research Laboratory (CGR) of the National Cancer Institute using the Infinium MethylationEPIC BeadChip (Illumina Inc.)[18].

## Bioinformatic analysis
All details of variant calling and filtering were identical to the previously published pipelines[18]. WGS processing is available in the public Terra workflow framework (Terra.bio)[85] REBC_methods_only. The WGS cohort included both cLNM-normal (blood or non-tumor thyroid tissue, as available) pairs and a reanalysis of the primary tumor (PT)-normal pairs to confirm the reproducibility of the variant calling pipeline, which resulted in 100% concordance in variant calls between the prior[18] and current analysis of the PT-normal pairs.

**Simple somatic variant detection and filtering pipeline.** A consensus calling approach for SSVs combined evidence from multiple detection algorithms[18]. Single nucleotide variants (SNVs) were detected by MuTect1.0 version GATK3 v1.1.6, MuTect2.0 version GATK3 "3.6-97-g881c5e9", Strelka1 version 1.0.11, and Strelka2 version 2.8.3. Insertion/deletion variants (indels) were detected by MuTect2.0, Strelka1, and Strelka2, as well as PCAWG_snowman version 1.0, and SvABA version 134. The mutations were filtered for possible sources of artifacts, including 8-oxoG damage, Panel of Normal evidence, local realignment issues, tumor sample contamination, alternate read support in the paired normal, and the post-process consensus filter requiring evidence from two independent algorithms. Two cLNM samples were excluded during the initial quality control evaluation, one because there was no overlap between the cLNM and PT SSVs, and the other because of contamination of the PT sample, resulting in a final analytic dataset of $N = 47$ cLNM samples. All samples are derived from unique individuals.

**Structural variant detection and filtering pipeline.** The SV detection pipeline was based on the consensus of calls among four structural variation algorithms: dRanger/Breakpointer[86], SvABA[87], PCAWG_snowman[88], and Manta[89]. SV variant calls from each algorithm were converted into a common format and filtered based on PCAWG PoN data. The SV post-process filters required variant evidence from at least two algorithms, at least four alternate allele supporting reads from the tumor sample, at most one alternate supporting read in the normal sample, and VAF ≥0.05. The SV filter also excluded SVs with breakpoints within centromere or telomere regions and within hotspot regions as previously described[18]. SV events were classified as simple/balanced, simple/unbalanced, and complex clusters of SV events based on the breakpoint proximity of the SV calls[90]. IGV manual review was performed for all fusion drivers, all simple/unbalanced SVs, and any complex SVs from individuals with multiple complex clusters. Any discordant events or false positives were corrected or removed to generate the final SV dataset.

**SCNA detection and filtering pipeline.** Copy number alterations were detected in the WGS data using the GATK 4.1.4 CNV workflows and multiple collapsing steps[18]. The SCNA PoN consisted of 423 CTB normal samples (blood and non-tumor tissue) with no sign of tumor-in-normal (TiN) contamination from the prior analysis[18]. Both total copy numbers based on normalized read coverage and germline heterozygous SNP allele fraction shifts were used to estimate the allelic copy number ratio across the genome and to assign allelic copy number ratios to discrete segments. The Fraction of Genome Altered was calculated by summing the span of each SCNA over the total span of the genome.

**Purity and ploidy assessment.** ABSOLUTE was utilized to estimate tumor ploidy. Given the limited allelic copy number signal in most samples, tumor purity was estimated considering both ABSOLUTE and a somatic VAF method based on the clonal allele fraction peak[90] as described previously[18]. For the purposes of most statistical analyses, we excluded cLNM and PT samples with tumor purity <20%. Figure S4 and Table S9 describe the availability of cLNM and PT samples by platform.

**Variant clonality (cancer cell fraction) estimation.** Since most tumors did not have sufficient copy numbers to constrain ABSOLUTE purity and ploidy solutions with corresponding allelic SCNAs and mutation CCFs, a custom approach was implemented to predict CCF based on the logic of the ABSOLUTE algorithm[18,90]. CCF Clonal and Subclonal Thresholds: For the SSV mutations, clonal variants were defined as those with a CCF hat ≥0.6, while subclonal SSV variants were those with a CCF hat <0.6.

**Mutational signature analysis.** Mutational signature classification was performed on single base substitutions (SBS) and small indel (ID) variants separately using SigProfiler[32,33]. The cLNMs and paired PTs had to be analyzed separately due to the substantial overlap in variants, therefore we classified signatures in two separate runs: (1) 45 high-purity cLNMs, 316 PTs from other individuals, and 42 samples from TCGA and (2) 41 high-purity paired PTs, and the same 316 PTs from other individuals and 42 samples from TCGA. Mutational signatures were predicted using SigProfilerMatrixGenerator (version 1.1.0), presig version 0.0.1, and SigProfilerExtractor version 1.0.3 applying non-negative matrix factorization-based signature extraction. In brief, SBS and ID mutations were optimally attributed and classified as mutational patterns of the known 96 SBS or 83 ID signatures from the Catalog of Somatic Mutations in Cancer (COSMIC v3)[33] or de novo signatures.

**mRNA clustering analyses.** Unsupervised consensus clustering analysis was performed using ConsensusClusterPlus[91]. Input data included the variance stabilizing transformation (vst) expression of the 1000 most variably expressed genes for mRNA analysis[92]. The appropriate cluster number was determined by identifying the largest cluster with a delta area value >0.3.

**PTC driver identification.** Candidate mutation and fusion drivers were identified in the 47 cLNM and reevaluated in the paired PT, examining both the WGS and mRNA-seq data with a comprehensive list of known driver genes[18]. Known mutation driver genes included genes significantly mutated in the prior MutSig2CV analysis[18], genes previously reported as mutation drivers in the TCGA analysis[93], and genes reported in the COSMIC Cancer Gene Census (CGC) database to version 92 (v92) (https://cancer.sanger.ac.uk/census)with frameshift, missense, nonsense, or splice site mutation types. Candidate driver mutations in the cLNM and PT tumors were somatic protein-altering variants that had a corresponding match in the COSMIC CGC v92 database within one of the known mutation driver genes.

The known fusion driver genes included previously identified fusion drivers or focally deleted genes in the prior analysis[18] or the TCGA analysis[93] and genes reported as oncogenes or tumor suppressors in the COSMIC CGC v92 with fusion indicated for either the "role in cancer" or "mutation type." Candidate fusion drivers in the cLNM and PT tumors were WGS structural variants and RNA-seq gene fusions involving one of the known fusion driver genes.

All 47 cLNM and PT pairs had the same candidate driver(s). For 45 pairs, a sole candidate driver was identified and therefore was designated as the final driver. For two pairs, two candidate drivers were identified, but the final driver was designated as the one that was recurrently altered in the full dataset of 487 (47 cLNM, 440 PT) samples.

**Methylation filtering.** The methylation filtering and normalization processing was updated in the current analysis compared with the prior analysis[18]. The methylation intensity files from the Illumina methylation assay on the MethylationEPIC BeadChip were processed with the R packages minfi and ChAMP[94]. Samples with mMed or uMed <10.2 were removed from the analysis prior to matching the cLNM with additional PT samples based on driver. All 165 samples in the analysis (43 cLNM, 36 paired PT, and 86 matched PT; Data S1) had a low fraction of failed probes (<0.005% failed CpG fraction). Probes were filtered at a detection $p$ value of 0.01 ($n = 25,429$), and due to a beadcount <3 in at least 5% of samples ($n = 1228$). Non-CpG probes ($n = 2804$), probes with SNPs[95] ($n = 53,375$), and probes aligning to multiple locations[96] ($n = 11$) were also filtered. After filtering, 783,071 probes were available for analysis. Estimates of the composition of cell types were obtained using the algorithm of ref. 97.

## Statistical analysis

All analyses were conducted using SAS version 9.4 (Cary, NC), R version 3.6.3 and 4.2.1 (Foundation for Statistical Computing, Vienna, Austria), or Epicure version 2.0 (Risk Sciences International, Ottawa, Canada).

**Predictors of cLNM occurrence.** We identified clinical and molecular predictors of cLNM occurrence in our primary study population of 440 adolescents and young adults with PTC using unadjusted logistic regression models and multivariate logistic regression models adjusted for sex and age at PTC (continuous). Additional sensitivity analyses were further adjusted for radiation dose (continuous), with doses >1000 mGy truncated (i.e., assigned the value of 1000 mGy) to reduce their influence on the estimated model coefficients[18]. For each predictor variable, statistical significance was assessed using a two-sided $P$ value generated using likelihood ratio tests, comparing model fit with and without the variable of interest. Models that evaluated the linear trend in radiation dose estimated the excess odds ratio (EOR) so that the effect of dose was linear (rather than log-linear), consistent with standard practice in radiation epidemiology[98].

**Genomic profiles of cLNM versus PT samples.** We compared the genomic profiles of cLNM versus PT samples using multivariable regression models adjusted for sex and age at PTC (continuous). Additional sensitivity analyses were further adjusted for radiation dose (continuous) and driver type (BRAF mutation, RET fusion, other RTK fusion, and BRAF fusion [referent group]). The type of regression model depended on the distribution of the molecular characteristic, as described previously[18] and specified in Table S12, with the use of linear regression models for continuous variables, logistic regression models for dichotomous characteristics, and proportional odds models for characteristics with discrete counts over a limited range.

**Transcriptomic profiles of cLNM versus PT samples.** To evaluate genes differentially expressed between cLNM versus PT samples, we

used standardized approaches as described by the Pancancer Analysis of Whole Genomes (PCAWG) Working Group[99]. Briefly, RNA-seq read counts for each gene were generated via STAR[100] and loaded into R version 4.1.2 (Foundation for Statistical Computing, Vienna, Austria). For mRNA-seq analyses, normalization size factors were calculated using the standard "Upper Quartile" method[101], in which each sample's size factor is equal to the 75th percentile of the set of nonzero gene-level read counts for that respective sample. The log-normalized counts are defined as:

$$Y_{i,g} = log_2\left[\frac{K_{i,g}+1}{s_i}\right] \qquad (1)$$

where $K_{i,g}$ is the simple read count for sample $i$ and gene $g$, $s_i$ is the normalization size factor for sample $i$, and $Y_{i,g}$ is the log-normalized read count for sample $i$ and gene $g$. Due to the large size of the sample set, specialized methods to calculate the dispersion were not necessary. For miRNA-seq analyses, we normalized read counts according to DESeq and excluded miRNAs with mean counts <1 (Supplementary Data 8).

As described above for analyses of the genomic profiles, simple linear regression models on the normalized read counts were used to determine whether tissue status (cLNM vs. PT) was associated with differential gene and miRNA expression, adjusted for sample batch, sex, and age at PTC, with additional sensitivity analyses adjusted for radiation dose and driver type, as above for the genomic profile analyses. Two-sided adjusted $p$ values were calculated using the standard Benjamini–Hochberg false discovery rate (FDR) method[102].

Patterns of mRNA-seq expression in previously identified gene sets were analyzed from the Molecular Signatures Database (MSigDB) "Hallmark gene sets" ($n = 50$; MSigDB v7.1; https://www.gsea-msigdb.org/gsea/msigdb)[45,46]. Expression information across gene sets was collapsed using GSVA, an R package that performs "Gene Set Variation Analysis," providing a Kolmogorov–Smirnov-like rank statistic based on the log-normalized counts for each gene and set of genes. Linear regression analyses were then performed on these statistics[103], as described above.

Finally, we used CIBERSORTx to estimate the cell type composition of cLNM and PT samples[49]. Single-cell RNA-seq data for thyroid cancer were downloaded from the Gene Expression Omnibus database (accession GSE184362)[50] to serve as training data. Clustering and cell type identification was performed in the Seurat R package[104], sorting the 169,161 cells into clusters of principal-components-based linear dimensional reduction. Cell types associated with each cluster were assigned based on reference transcript lists from the thyroid from The Human Protein Atlas as well as the original thyroid cancer single-cell RNA-seq data[105–107]. Cell fractions for each cLNM and PT sample were then imputed using CIBERSORTx[49].

**Epigenomic profiles of cLNM versus PT samples.** To evaluate differential DNA methylation between cLNM versus PT samples, outliers were fixed by replacing all values ≤0 with the smallest positive value and all values ≥1 with the largest value below 1. Normalization was performed by applying the Beta-Mixture Quantile (BMIQ) normalization with ChAMP.norm. ChAMP.SVD was utilized to analyze potential sources of technical or other variation. ChAMP.runComBat was used to account for variation due to sex and the array or Sentrix position.

Notably, all the cLNM samples were run on the same sample plate, separately from the PT samples, thus caution is warranted in the interpretation of our results. Sample plate was included as a covariate in the simple linear regression models used for analysis, which also were adjusted for sex and age at PTC, with additional sensitivity

analyses adjusted for radiation dose and driver type, as above for the genomic profile analyses. We also evaluated robust linear models with sandwich estimators for top sites, to account for heteroskedasticity.

### Reporting summary

Further information on research design is available in the Nature Portfolio Reporting Summary linked to this article.

## Data availability

Raw WGS, RNA-seq, and miRNA-seq data are deposited at the Genomic Data Commons under project ID REBC-THYR. These data, as well as raw SNP array data, were accessed through the database of Genotypes and Phenotypes (dbGaP) under accession code phs001134 [https://www.ncbi.nlm.nih.gov/projects/gap/cgi-bin/study.cgi?study_id=phs001134.v2.p1]. Data were available under dbGaP-controlled access for general research use. Approved users will be granted access to data for 12-months, after which the requestor will be asked to renew or close-out the project. The sample IDs included in each analysis are specified in Supplementary Data 1. Source data are provided with this paper, with specific sources noted in each figure and table in Supplementary Data 2–8. Processed scRNA-seq data from ref. 50 are available without restriction from the Gene Expression Omnibus (GEO) database under accession code GSE184362.

## Code availability

The Methods text specifies code that has been posted to GitHub and is archived on Zenodo[83,84,90,92]. Code used for mRNA-seq quality control and alignment is available at https://github.com/NCI-CGR/ChernobylThyroidCancer-RNAseq. Code for mRNA clustering analyses is available at https://github.com/NCI-CGR/ChernobylThyroidCancer-Clustering. Code for miRNA-seq processing is available at https://github.com/NCI-CGR/Gencode_microRNA-seq. Code for tumor purity and clonality estimation and SV events classification, is available at https://github.com/getzlab/REBC_tools/releases/tag/v1.1.2.

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

## Acknowledgements

The authors gratefully acknowledge the commitment of the staff of the Laboratory of Morphology of Endocrine System and the staff of the Department of Surgery of Endocrine System of the V.P. Komisarenko Institute of Endocrinology and Metabolism of the National Academy of Medical Sciences of Ukraine, who prepared the pathological material for the study and operated on the patients; the confirmation of thyroid tumor diagnoses provided by the International Pathology Panel of the Chornobyl Tissue Bank, including Professors A. Abrosimov, T. Bogdanova, G. Fadda, J. Hunt, M. Ito, V. Livolsi, J. Rosai, and E.D. Williams, and Dr. N. Dvinskyh; and Nathan Appel (Information Management Services, Inc., Calverton, MD) for programming support. This project was supported by the Intramural Program of the National Cancer Institute and the National Institutes of Health. The Chornobyl Tissue Bank was supported by the National Cancer Institute (U24CA082102), the European Commission, the Sasakawa Memorial Health Foundation of Japan (SMHF), and the World Health Organization (WHO). This work utilized the computational resources of the NIH HPC Biowulf cluster (http://hpc.nih.gov). The opinions expressed by the authors are their own, and this material should not be interpreted as representing the official viewpoint of the U.S. Department of Health and Human Services, the National Institutes of Health, or the National Cancer Institute.

## Author contributions

L.M.M., O.W.L., D.M.K., T.I.B., M.D.T., G.A.T., and S.J.C. conceived the project. T.I.B., V.D., S.M., M.C., L.Y.Z., M.K., M.D.T., and G.A.T. collected the samples and radiation exposure data. L.M.M., O.W.L., D.M.K., C.S., S.W.H., C.E.B., S.J.S., J.D., M.Y., A.H., B.D.H., C.L.D., M.K.S., Kr.J., Ko.J., B.J., E.T.D., J.M.G., and J.B. processed the samples and/or resulting laboratory data. L.M.M., O.W.L., D.M.K., S.W.H., C.E.B., and S.J.S.

conducted analyses. L.M.M., O.W.L., D.M.K., and S.J.C. drafted the paper. All authors (L.M.M., O.W.L., D.M.K., T.I.B., C.S., S.W.H., C.E.B., S.J.S., E.K.C., V.D., S.M., M.C., L.Y.Z., J.D., M.K., M.Y., A.H., B.D.H., C.L.D., M.K.S., Kr.J., Ko.J., B.J., M.J.M., E.T.D., V.V., J.M.G., J.B., K.M., M.H., A.B.G., G.G., M.D.T., G.A.T., and S.J.C.) helped to interpret the data and read, revised, and approved the final manuscript.

## Funding

## Competing interests

E.T.D. is an employee of Nvidia Corporation and owns stock in Nvidia, Illumina, and Pacific Biosciences. G.G. receives research funds from IBM and Pharmacyclics, and is an inventor on patent applications related to MuTect, ABSOLUTE, MutSig, MSMuTect, MSMutSig, MSIdetect, POLYSOLVER, and TensorQTL. G.G. is a founder, consultant and holds privately held equity in Scorpion Therapeutics. The remaining authors declare no competing interests.

## Additional information

[1]Radiation Epidemiology Branch, Division of Cancer Epidemiology and Genetics, National Cancer Institute, National Institutes of Health, Bethesda, MD, USA. [2]Laboratory of Genetic Susceptibility, Division of Cancer Epidemiology and Genetics, National Cancer Institute, National Institutes of Health, Bethesda, MD, USA. [3]Laboratory of Morphology of the Endocrine System, V.P. Komisarenko Institute of Endocrinology and Metabolism of the National Academy of Medical Sciences of Ukraine, Kyiv, Ukraine. [4]Broad Institute of MIT and Harvard, Cambridge, MA, USA. [5]Occupational and Environmental Epidemiology Branch, Division of Cancer Epidemiology and Genetics, National Cancer Institute, National Institutes of Health, Bethesda, MD, USA. [6]National Research Center for Radiation Medicine of the National Academy of Medical Sciences of Ukraine, Kyiv, Ukraine. [7]Cancer Genomics Research Laboratory, Leidos Biomedical Research Inc., Frederick National Laboratory for Cancer Research, Bethesda, MD, USA. [8]Department of Surgery and Cancer, Imperial College London, Charing

Cross Hospital, London, United Kingdom. [9]Integrative Tumor Epidemiology Branch, Division of Cancer Epidemiology and Genetics, National Cancer Institute, National Institutes of Health, Bethesda, MD, USA. [10]Nvidia Corporation, Santa Clara, CA, USA. [11]Nationwide Children's Hospital, Biospecimen Core Resource, Columbus, OH, USA. [12]Departments of Pathology and Pediatrics, Ohio State University College of Medicine, Columbus, OH, USA. [13]Center for Cancer Research and Department of Pathology, Massachusetts General Hospital, Boston, MA, USA. [14]Harvard Medical School, Boston, MA, USA. [15]Department of Fundamental and Applied Problems of Endocrinology, V.P. Komisarenko Institute of Endocrinology and Metabolism of the National Academy of Medical Sciences of Ukraine, Kyiv, Ukraine. [16]These authors contributed equally: Lindsay M. Morton, Olivia W. Lee, Danielle M. Karyadi, Tetiana I. Bogdanova. [17]These authors jointly supervised this work: Mykola D. Tronko, Gerry A. Thomas, Stephen J. Chanock. ✉e-mail: mortonli@mail.nih.gov; chanocks@mail.nih.gov

