## [Peer Review File · Nature Communications]

REVIEWER COMMENTS

Reviewer #1 (Remarks to the Author): Expert in PTC genomics, therapy, and PTC-cLNMs

Based on their prior research in Science. 2021 May 14;372(6543), the authors further evaluated data from different cohorts and TCGA and primarily discovered that fusion-driver had a greater cLNM metastatic frequency than BRAF mutation driver. This extensive research of PTC caused by radiation has significant ramifications.

There are several major comments as follows.

1. cLNM includes patients with N1a and N1b, while patients with N1b require lateral cervical lymph node dissection. Is there a difference between N1a and N1b when comparing the effects of the Fusion- and Mutation- driver on the cLNM?
2. Is there any additional pathology-based subtyping of PTC? The PTC subtypes have been further divided in recent years, according to WHO (2022), which may not be the same as the surgical pathology at that time. Reconfirming the pathology might be difficult, but if authors can make this part of the data better, researchers might be able to identify the subtype of radiation-associated fusion driver.
3. Some patients with tumors smaller than 4 cm were labeled as T3 in SupplementaryFigure 1, which might be because postoperative pathology identified extra-glandular invasion of the tumor. If possible, can authors please add the location of the extraglandular invasion? That will help surgeons and pathologists pay more attention in clinical practice.

And several minor comments.

1. For TNM staging, the authors used the AJCC,2010,7Edition. It is advised that the authors update the reference version of TNM staging even though the definition of T staging in the AJCC 8th edition is essentially unchanged.
2. The authors showed in sFig8 and sFig9 that the differences in transcriptome were not caused by variations in cellular composition. However, using a deconvolution technique like Cibersort to determine the percentage of lymphocytes in the samples may be more understandable. Additionally, the Tabula Sapiens Consortium single cell data for immune cells in healthy populations and immune cells linked to cancer may not be very reliable. Use of the currently published Nat Commun. 2021 Oct 18;12(1):6058 or Sci Adv. 2021 Jul 28;7(31) data is advised if validation at single-cell resolution is required.

Reviewer #2 (Remarks to the Author): Clinical expert in PTC genetics and genomics

In this manuscript the authors explored an association between cervical lymph node metastases (cLNM) occurrence and a range of patient, clinical, epidemiologic, and molecular characteristics using genomic landscape data of 440 patients (81 131I-unexposed and 359 131I exposure before adulthood from the Chernobyl accident). They found that cLNM were more frequent in PTC with fusion versus mutation drivers.

This original study is a high quality and noteworthy manuscript to the thyroid field, as it reveals that age and radiation exposure are not related to cLNM in patients predominantly diagnosed during young adulthood and suggested that the PTC driver is the dominant factor associated with cLNM, with the highest frequency of cLNM in tumors with RET or NTRK1 fusion drivers. By reanalyzing previously published data from two Chernobyl studies with available 131I exposure data and The TCGA data, restricting to those with known fusion or mutation drivers, the author found that cLNM occurrence was related to PTC driver.

Moreover, transcriptomic analysis of cLNM and Primary tumor (44 paired samples) showed that three genes centered on the HOXC cluster on chromosome 12q13.13 were differentially expressed. After adjusting for HOXC10 expression, none of the remaining genes in the HOXC locus were significant. Further, the authors suggested the mechanism associated with HOXC locus dysregulation is unlikely due to somatic copy number alteration (SCNAs) or epigenomic differences, providing direction for future research on the biological underpinnings of PTC cLNM.

Overall, this is a clear and well-written manuscript. The authors provided information about their previous study results which allow the readers track the study rationale and novelty. The results are solid and support their conclusions. Although the methods used were clear and precise, clarification of few points could be provided.

Comments:

1) Although the authors suggested that mechanism associated HOXC locus dysregulation is unlikely somatic copy number alteration (SCNAs) and epigenomic differences, only DNA methylation profiling was performing. Even if DNA methylation has been the main mechanism association with expression of these genes, a different epigenetic mechanism such as histone modification or lncRNA or miRNA may be linked to HOX gene expression. The use of an alternative promoter was previously associated with histone modification. Recently, few studies demonstrated that HOX gene expressions might be influenced by variety of hormones that include estradiol, progesterone, testosterone. Hence, did the authors found any correlation with miRNA-profiling data?

2) Table 1 showed that most of tumors with cLNM showed BRAF mutation (73 cases) than RET fusion (52 cases). Very few cases (10 cases) were available for the differential expression analysis of PTC and cLNM performed in 44 paired samples, while the remaining cases presented fusions (18 cases with RET fusion). Could this "bias" interfere in the expression profiling and signature found?

3) Remarkably, none of the cases showed distant metastasis. Can the authors comment on that.

4) The authors did not comment their results on relative telomere length quantification.

5) please double check the typos on the word Chernobyl

Reviewer #3 (Remarks to the Author): Expert in radiation oncology and genomics

The paper reports on a study expansion of molecular profiling of a post-Chernobyl CTB cohort of papillary thyroid carcinoma (PTC) published previously in Science (doi:10.1126/science.abg2538) with a

special focus on cervical lymph node metastases (cLNM). The authors further present a comprehensive genomic landscape of 47 cLNM samples in comparison to matched primary tumors (PT).

While I explicitly acknowledge the importance of such comprehensive data from large tumor cohorts and in particular matched PT-cLNM tumor pairs, there are some major concerns about potential bias, incomplete evaluation and data interpretation.

1. The study population of 440 PTC had obviously 164 cLNM which were used for multivariable modeling approaches and stratification to unravel various effects of clinical, epidemiologic and molecular features. However, only n=49 (according to Supplementary Methods, n=47 according to the Abstract) cLNM were analyzed in comparison to molecular PT data. At this point the study design is not clear because detailed information about this subgroup (age at PTC or radioiodine exposure) was not disclosed. It is not clear if multiple cLNM from one individual patient are included. Several earlier epidemiologic and molecular studies on radiation-associated PTCs including CTB samples suggested convincingly different radiation effects in age-related subgroups of post-Chernobyl PTC. Such considerations are completely missing in the present paper, i.e. if a younger subgroup of the study population shows opposite correlations than an older subgroup. In this sense it is important to disclose the allegiance of the analyzed cLNM-PT tumor pairs to a particular subgroup. In this context the statement on page 8/line 129 "...there was no measurable effect of cumulative environmental radiation exposure on cLNM occurrence" must be revisited.

2. According to age-related subgroups of post-Chernobyl PTC as mentioned above, age stratification as outlined on page 8/line 140 ("...analyses stratified by age at PTC (<30 vs. ≥30 years) revealed that the frequency of cLNM occurrence was consistent by age in tumors with RET 141 fusion drivers") is incomprehensible because a rationale for an age threshold at 30 years is missing and cannot be derived from any epidemiologic study on post-Chernobyl PTC published so far. Here, subcohort-specific characteristics can act as major obstacle disentangling independent effects e.g., due to markedly different estimates of the proportion of radiation-induced PTCs within exposed cases (range from 55% for UkrAm cases to 85% in other CTB cohorts depending on mean attained age and mean dose). Therefore, a re-analysis of data within established epidemiologic age-related subgroups of post-Chernobyl PTC is necessary.

3. I am missing a detailed differential gene expression analysis between metastasizing and non-metastasizing PT or between cLNM and PT pairs focusing on enriched gene sets and activity scores of metastasis-related gene signatures (e.g., p-EMT or EMT signatures). Apart from the discussion about fusion and mutational drivers, this would gain substantial information about molecular metastatic processes. It must be assumed that the postulated higher aggressiveness of fusion drivers in terms of cLNM formation should involve processes of epithelial-mesenchymal-transition which is a hallmark of tumor metastasis.

4. In this context, the applied RNAseq data analysis might introduce substantial bias within differential expression analysis and derived results and interpretation. There is no need to deviate from the well-established and accessible methods like DESeq2 and normalization methods that account for e.g. sequencing depth and other technical biases specific to RNAseq data. Details and rationales on e.g. definition of differential gene expression seem arbitrary and are not using commonly accepted thresholds in RNAseq studies. I suggest a reanalysis of the transcriptomic data using state-of-the-art bioinformatic approaches.

5. The bottom line message of this paper is a higher metastatic potential of fusion drivers in a post-Chernobyl PTC cohort of young patients. I am missing a discussion on potential mechanisms how fusion vs. mutational drivers can impact on the metastatic process. If the paradigm of a multi-step process of tumor progression is true, additional molecular mechanisms must be initiated to enter tumor metastasis. A more detailed discussion on this aspect is needed.

6. Figure 1 depicts a share cLNM yes/no of 40/60 for BRAF and 70/30 for RET with an OR 3.5 which supports the initial hypothesis. However, the groups Mut (other) vs. Fusion (other) do not reveal differential enrichment of cLNM. Table 1 summarizes the results of multivariate logistic regression on cLNM yes/no separately for BRAF and RET with mostly insignificant ORs, thus not suggesting differential metastasizing potential. Therefore, additional statistical analysis is needed to include multivariate logistic regression based on the whole data set used for Figure 1 and the covariables of Table 1. Regression analysis should be followed by AUC calculations with and without cross validation (AUC > 0.8 without cross validation and > 0.7 with cross validation are deemed necessary to support initial hypothesis).

7. Page16/line 309-314: The reasoning here is somewhat puzzling. The previous Science publication (doi:10.1126/science.abg2538) reports a robust dose response of the driver type marker. In the present study the driver type marker is associated with cLNM prevalence which does not exhibit a dose response. How can both statements be reconciled?

Reviewer #1 (Remarks to the Author): Expert in PTC genomics, therapy, and PTC-cLNMs Based on their prior research in Science. 2021 May 14;372(6543), the authors further evaluated data from different cohorts and TCGA and primarily discovered that fusion-driver had a greater cLNM metastatic frequency than BRAF mutation driver. This extensive research of PTC caused by radiation has significant ramifications.

There are several major comments as follows.

1. cLNM includes patients with N1a and N1b, while patients with N1b require lateral cervical lymph node dissection. Is there a difference between N1a and N1b when comparing the effects of the Fusion- and Mutation- driver on the cLNM?

Response: We appreciate the reviewer raising this question. We analyzed our data separately for N1a and N1b and have added new text in the Results and Discussion as well as new data in Supplementary Tables S1 and S5 to describe our observations:

Page 7 and Table S1 (in the version of the manuscript with tracked changes): “Among patients with cLNM, approximately half were N1a (N=87, 48.9%) and half N1b (N=91, 51.1%).”

Pages 8-9 and Table S5: “The excess occurrence of cLNM associated with fusion drivers was more pronounced for N1b than N1a cLNM (N1b: fusion driver N=57/156, 32.4%, mutation driver N=32/252, 12.7%; N1a: fusion driver N=40/156, 22.7%, mutation driver N=44/252, 17.5%), particularly for NCOA4-RET (N1b: N=9/15, 60.0%; N1a: N=3/15, 20.0%) (Table S5).”

Page 17: “The slightly higher frequency of cLNM, particularly N1b, that we observed for tumors with *NCOA4-RET* than *CCDC6-RET* fusion drivers, albeit based on small numbers, also is consistent with previous reports^{53,54} of increased aggressiveness of tumors with *RET/PTC3* rearrangements.”

2. Is there any additional pathology-based subtyping of PTC? The PTC subtypes have been further divided in recent years, according to WHO (2022), which may not be the same as the surgical pathology at that time. Reconfirming the pathology might be difficult, but if authors can make this part of the data better, researchers might be able to identify the subtype of radiation-associated fusion driver.

Response: We agree that pathology data for PTC are very important. All tumors in this study underwent centralized pathology review by an international panel of at least five experts at the time of reporting to the Chernobyl Tissue Bank. Undertaking a comprehensive pathology review of the historical tumors in Kyiv at this time according the current 2022 WHO classification is beyond the scope of this analysis but is a priority for the future.

3. Some patients with tumors smaller than 4 cm were labeled as T3 in SupplementaryFigure 1, which might be because postoperative pathology identified extra-glandular invasion of the tumor. If possible, can authors please add the location of the extraglandular invasion? That will

help surgeons and pathologists pay more attention in clinical practice.

Response: We have added these data as requested in the Results:

Page 7: “Nearly half of the tumors (N=206, 46.8%) were classified as pathologic T1, 71 (16.1%) T2, and 163 (37.0%) T3 according to the 7th edition of TNM staging (Figure S1, Tables S1-S2)³¹. Of the N=163 tumors classified as T3, 106 tumors of any size had evidence of minimal extrathyroidal extension in the fat and connective tissue, 24 had evidence of extrathyroidal extension in the muscle, and 33 had no evidence of extrathyroidal extension but were sized >4 cm.”

And several minor comments.

1. For TNM staging, the authors used the AJCC,2010,7Edition. It is advised that the authors update the reference version of TNM staging even though the definition of T staging in the AJCC 8th edition is essentially unchanged.

Response: In preparing this study, the lead pathologist (T.I.B.) re-classified the tumors according to the 8th edition of TNM staging. As expected, the detection of cLNM occurrence was unchanged, but some of the tumors with minimal extrathyroidal extension in the fat and connective tissue were re-classified according to the tumor size. We have added this information to the Supplementary Methods:

Page 1: “For reference, when the PTCs were staged according to the TNM 8th edition⁹, tumors with minimal extrathyroidal extension in the fat and connective tissue that were classified as T3 in the 7th edition were re-classified according to tumor size. Specifically, 71 PTCs shifted from T3 to T1 and 34 from T3 to T2. However, we retained our primary analysis using the TNM 7th edition classification because we analyze tumor size separately and to enable comparison of our results with two previous studies in the literature¹⁰⁻¹², both of which were classified using the 7th edition.”

2. The authors showed in sFig8 and sFig9 that the differences in transcriptome were not caused by variations in cellular composition. However, using a deconvolution technique like Cibersort to determine the percentage of lymphocytes in the samples may be more understandable. Additionally, the Tabula Sapiens Consortium single cell data for immune cells in healthy populations and immune cells linked to cancer may not be very reliable. Use of the currently published Nat Commun. 2021 Oct 18;12(1):6058 or Sci Adv. 2021 Jul 28;7(31) data is advised if validation at single-cell resolution is required.

Response: We have expanded our analysis of cellular composition to include the deconvolution technique CIBERSORTx, utilizing data from Pu et al., *Nat Commun* 2021 as training data, as suggested by the reviewer. Reassuringly, these results also demonstrated comparable distributions in immune cell composition between cLNM and PT samples. The results from this analysis have been added to the manuscript as follows:

Page 15: “Additionally, application of CIBERSORTx to identify cell types similarly suggested comparable immune cell distributions in cLNM and PT samples (Figure S10)^{49,50}.”

Supplementary Methods: “Finally, we used CIBERSORTx to estimate the cell type composition of cLNM and PT samples³⁹. Single-cell RNA-seq data for thyroid cancer were downloaded from the Gene Expression Omnibus database (accession GSE184362)⁴⁰ to serve as training data. Clustering and cell type identification was performed in the Seurat R package⁴¹, sorting the 169,161 cells into clusters principal-components-based linear dimensional reduction. Cell types associated with each cluster were assigned based on reference transcript lists from the thyroid from The Human Protein Atlas as well as the original thyroid cancer single-cell RNA-seq data⁴²⁻⁴⁴. Cell fractions for each cLNM and PT sample were then imputed using CIBERSORTx³⁹.”

Reviewer #2 (Remarks to the Author): Clinical expert in PTC genetics and genomics

In this manuscript the authors explored an association between cervical lymph node metastases (cLNM) occurrence and a range of patient, clinical, epidemiologic, and molecular characteristics using genomic landscape data of 440 patients (81 131I-unexposed and 359 131I exposure before adulthood from the Chernobyl accident). They found that cLNM were more frequent in PTC with fusion versus mutation drivers.

This original study is a high quality and noteworthy manuscript to the thyroid field, as it reveal that age and radiation exposure are not related to cLNM in patients predominantly diagnosed during young adulthood and suggested that the PTC driver is the dominant factor associated with cLNM, with the highest frequency of cLNM in tumors with RET or NTRK1 fusion drivers. By reanalyzing previously published data from two Chernobyl studies with available 131I exposure data and The TCGA data, restricting to those with known fusion or mutation drivers, the author found that f cLNM occurrence was related to PTC driver.

Moreover, transcriptomic analysis of cLNM and Primary tumor (44 paired samples) showed that three genes centered on the HOXC cluster on chromosome 12q13.13 were differentially expressed. After adjusting for HOXC10 expression, none of the remaining genes in the HOXC locus were significant. Further, the authors suggested the mechanism associated with HOXC locus dysregulation is unlikely due to somatic copy number alteration (SCNAs) or epigenomic differences, providing direction for future research on the biological underpinnings of PTC cLNM.

Overall, this is a clear and well-written manuscript. The authors provided information about their previous study results which allow the readers track the study rationale and novelty. The results are solid and support their conclusions. Although the methods used were clear and precise, clarification of few points could be provided.

Comments:

1) Although the authors suggested that mechanism associated HOXC locus dysregulation is unlikely somatic copy number alteration (SCNAs) and epigenomic differences, only DNA methylation profiling was performing. Even if DNA methylation has been the main mechanism association with expression of these genes, a different epigenetic mechanism such as histone modification or lncRNA or miRNA may be linked to HOX gene expression. The use of an alternative promoter was previously associated with histone modification. Recently, few studies

demonstrated that *HOX* gene expressions might be influenced by variety of hormones that include estradiol, progesterone, testosterone. Hence, did the authors found any correlation with miRNA-profiling data?

Response: We appreciate the opportunity to conduct miRNA-seq, expanding the scope of our comparison of the molecular profiles of cLNM and primary tumor samples in response to this Reviewer comment. Details of the laboratory methods, bioinformatics, and statistical analysis were added to the Supplementary Methods, and sample counts were added to Figure S4 and Table S9.

Our analyses revealed four miRNAs that were strikingly differentially expressed between cLNM and primary tumor samples, including two that are also in the *HOXC* locus, supporting our mRNA-seq analyses that already identified dysregulation of this locus in cLNM. We describe these new findings in the revised Abstract, Results, and Discussion, as well as adding the key results in a new Figure (Figure 5) and the full modeling results in Table S16:

Abstract, Page 3: “the strongest differentially expressed microRNA also was near *HOXC10* ($P_{\text{miR-196a2}}=1.3\times 10^{-25}$).”

Results, Page 15: “In parallel analyses of miRNA, we identified four miRNAs that were substantially differentially expressed: overexpression of miR-196a2 ($\log_2\text{FC}=3.9$, $P_{\text{adj}}=1.3\times 10^{-25}$) and miR-615 ($\log_2\text{FC}=2.3$, $P_{\text{adj}}=3.2\times 10^{-18}$) and underexpression of miR-137 ($\log_2\text{FC}=-2.5$, $P_{\text{adj}}=7.7\times 10^{-13}$) and miR-141 ($\log_2\text{FC}=-1.0$, $P_{\text{adj}}=1.6\times 10^{-12}$) in cLNM compared with PT samples (Figure 5A-E, Table S16. miR-196a2 and miR-615 have correlated expression ($r=0.67$) and are located in the *HOXC* locus near *HOXC10* and *HOXC5*, respectively. Further adjustment of models for radiation dose and driver as well as restriction of analyses to the cLNM and paired PT samples from the same individuals generally yielded similar results (Table S16).”

Discussion, Page 17: “...striking transcriptomic changes centered on the *HOXC* cluster on chromosome 12q13.13, as supported by both mRNA-seq and miRNA-seq results, provide direction for future research on the biological underpinnings of PTC cLNM.”

Discussion, Page 18: “In transcriptomic analysis of paired PT and cLNM, we identified a number differentially expressed genes and miRNAs. Most notably, we observed overexpression in cLNM of *HOXC10*, *HOTAIR*, and 7 other genes in the 12q13.13 *HOXC* locus, as well as two miRNAs in the same locus (miR-196a2 and miR-615).”

Discussion, Pages 19-20: “These associations are further supported by our observation of reduced expression of miR-137 and miR-141, both of which are tumor suppressor miRNAs that reportedly play a role in cancer occurrence and progression^{70,71}. miR-141 in particular is part of the miR-200 family, which has been shown to target and inhibit the ZEB1 and ZEB2 EMT transcription factors and is well described in cancer metastasis^{71,72}; miR-141 also specifically has been shown to be downregulated in thyroid

cancer, with correlations between expression and cellular proliferation, apoptosis, and migration⁷³.”

2) Table 1 showed that most of tumors with cLNM showed BRAF mutation (73 cases) than RET fusion (52 cases). Very few cases (10 cases) were available for the differential expression analysis of PTC and cLNM performed in 44 paired samples, while the remaining cases presented fusions (18 cases with RET fusion). Could this “bias” interfere in the expression profiling and signature found?

Response: In Table S9, we provide information on the distribution of unavailable vs. included cLNM samples by PTC driver. To address the reviewer’s question, we have created Table S10, which provides data demonstrating that sample inclusion was not related to patient sex, age, or prior radiation exposure, as described in the following Results text:

Page 11: “Although the available cLNM samples had a higher proportion of fusion-driven tumors (Table S9), the distributions with respect to patient age, sex, and radiation dose were otherwise comparable (Table S10).”

3) Remarkably, none of the cases showed distant metastasis. Can the authors comment on that.

Response: We identified 14 tumors with both cLNM and distant metastases and 1 with distant metastases only (Results Page 7, Tables S1-S2). We note in the Discussion (Page 20) that, “The paucity of distant metastases was notable in our study population, though not all patients were systematically scanned for distant events.”

4) The authors did not comment their results on relative telomere length quantification.

Response: We appreciate the opportunity to conduct relative telomere length quantification, expanding the scope of our comparison of the molecular profiles of cLNM and primary tumor samples in response to this Reviewer comment. Details of the laboratory methods and bioinformatics were added to the Supplementary Methods, and sample counts were added to Figure S4 and Table S9. We did not find a substantial difference in the relative telomere length between cLNM and primary tumor samples, as described in Table S10 and in text added to the Results:

Pages 12-13: “In multivariable regression models adjusted for age and sex, the distribution of the measured genomic characteristics was not statistically different between cLNM and PT samples, except the relative telomere length was suggestively shorter in cLNM (P=0.014) (Table S12). Results generally were comparable when we further adjusted our initial models for radiation dose and driver, and when we restricted analyses to the paired cLNM-PT samples (N=41) from the same individuals (Figure S5, Table S12), except the suggestively shorter relative telomere length in cLNM was no longer evident in the paired analysis (P=0.24).”

5) please double check the typos on the word Chornobyl

Response: We have utilized the Ukrainian spelling “Chornobyl” in all written text throughout the manuscript. We note, however, that many previous publications use the Russian-derived spelling “Chernobyl,” and our reference section retains the spelling utilized in each publication.

Reviewer #3 (Remarks to the Author): Expert in radiation oncology and genomics

The paper reports on a study expansion of molecular profiling of a post-Chernobyl CTB cohort of papillary thyroid carcinoma (PTC) published previously in Science (doi:10.1126/science.abg2538) with a special focus on cervical lymph node metastases (cLNM). The authors further present a comprehensive genomic landscape of 47 cLNM samples in comparison to matched primary tumors (PT).

While I explicitly acknowledge the importance of such comprehensive data from large tumor cohorts and in particular matched PT-cLNM tumor pairs, there are some major concerns about potential bias, incomplete evaluation and data interpretation.

1. The study population of 440 PTC had obviously 164 cLNM which were used for multivariable modeling approaches and stratification to unravel various effects of clinical, epidemiologic and molecular features. However, only n=49 (according to Supplementary Methods, n=47 according to the Abstract) cLNM were analyzed in comparison to molecular PT data. At this point the study design is not clear because detailed information about this subgroup (age at PTC or radioiodine exposure) was not disclosed. It is not clear if multiple cLNM from one individual patient are included. Several earlier epidemiologic and molecular studies on radiation-associated PTCs including CTB samples suggested convincingly different radiation effects in age-related subgroups of post-Chernobyl PTC. Such considerations are completely missing in the present paper, i.e. if a younger subgroup of the study population shows opposite correlations than an older subgroup. In this sense it is important to disclose the allegiance of the analyzed cLNM-PT tumor pairs to a particular subgroup. In this context the statement on page 8/line 129 “...there was no measurable effect of cumulative environmental radiation exposure on cLNM occurrence” must be revisited.

Response: We appreciate the importance of providing further information on sample inclusion, as noted in this comment and also by Reviewer #2. Our final analytic dataset included N=47 cLNM samples. Although N=49 samples were originally identified, we have further clarified the exclusion of two samples in the Supplementary Methods (Pages 2-3). Additionally, we have noted that only one cLNM from each individual patient was included:

Supplementary Methods, Pages 2-3: “Two cLNM samples were excluded during initial quality control evaluation, one because there was no overlap between the cLNM and PT SSVs, and the other because of contamination of the PT sample, resulting in a final analytic dataset of N=47 cLNM samples. All samples derived from unique individuals.”

As noted above, we have created Table S10, which provides data demonstrating that sample inclusion was not related to patient sex, age, or prior radiation exposure:

Page 11: “Although the available cLNM samples had a higher proportion of fusion-driven tumors (Table S9), the distributions with respect to patient age, sex, and radiation dose were otherwise comparable (Table S10).”

Finally, we appreciate the opportunity to clarify that the statement on Page 8 that “there was no measurable effect of cumulative environmental radiation exposure on cLNM occurrence ($P_{\text{trend}}=0.32$)” and on Page 9 that “No dose-response trend was observed between radiation and cLNM in categorical, linear, linear-quadratic, or quadratic models.” refer to analyses that were conducted in our full study population and thus were not dependent on cLNM sample availability.

2. According to age-related subgroups of post-Chernobyl PTC as mentioned above, age stratification as outlined on page 8/line 140 (“...analyses stratified by age at PTC (<30 vs. ≥30 years) revealed that the frequency of cLNM occurrence was consistent by age in tumors with RET 141 fusion drivers”) is incomprehensible because a rationale for an age threshold at 30 years is missing and cannot be derived from any epidemiologic study on post-Chernobyl PTC published so far. Here, subcohort-specific characteristics can act as major obstacle disentangling independent effects e.g., due to markedly different estimates of the proportion of radiation-induced PTCs within exposed cases (range from 55% for UkrAm cases to 85% in other CTB cohorts depending on mean attained age and mean dose). Therefore, a re-analysis of data within established epidemiologic age-related subgroups of post-Chernobyl PTC is necessary.

Response: We have clarified in the Results (Page 9) and in a new footnote added to Figure S2 that we used 30 years as a cutpoint, “reflecting the mean age at PTC diagnosis among exposed individuals.” In response to this reviewer comment, we have added a new Supplementary Table (Table S6) that provides more detailed breakdowns of cLNM occurrence by driver and age at PTC, as noted in new Results text:

Page 9: “More detailed breakdowns of cLNM occurrence by driver and age at PTC are provided in Table S6.”

3. I am missing a detailed differential gene expression analysis between metastasizing and non-metastasizing PT or between cLNM and PT pairs focusing on enriched gene sets and activity scores of metastasis-related gene signatures (e.g., p-EMT or EMT signatures). Apart from the discussion about fusion and mutational drivers, this would gain substantial information about molecular metastatic processes. It must be assumed that the postulated higher aggressiveness of fusion drivers in terms of cLNM formation should involve processes of epithelial-mesenchymal-transition which is a hallmark of tumor metastasis.

Response: We conducted two new analyses to respond to this Reviewer comment. First, we evaluated differentially expressed genes between metastasizing and non-metastasizing PTC, but found no significant differences. These new analyses are described in the Results:

Page 10: “We further compared transcriptomic profiles of PT with and without cLNM but did not find any significantly differentially expressed genes (DEGs) in analyses restricted to the two most commonly occurring drivers, *BRAF* mutation and *RET* fusion (data not shown).”

Second, we expanded our evaluation of differentially expressed genes between cLNM and PT using gene-set enrichment analyses based on the Molecular Signatures Database (MSigDB) “Hallmark gene sets”, which specifically includes EMT, as described in the Supplementary Methods text. None of these signatures demonstrated major differences between cLNM and PT, as described in the Results and Discussion:

Supplementary Methods: “Patterns of mRNA-seq expression in previously identified gene sets were analyzed from the Molecular Signatures Database (MSigDB) “Hallmark gene sets” (n=50; MSigDB v7.1; <https://www.gsea-msigdb.org/gsea/msigdb>)^{36,37}. Expression information across gene sets was collapsed using GSVA, an R package that performs “Gene Set Variation Analysis,” providing a Kolmogorov-Smirnov-like rank statistic based on the log-normalized counts for each gene and set of genes. Linear regression analyses were then performed on these statistics³⁸, as described above.”

Results, Page 14: “Further analyses of gene expression using the Molecular Signatures Database (MSigDB) “Hallmark gene sets”^{45,46} did not yield any statistically significant differences between cLNM and PT (Table S15).”

Discussion, Page 19: “*HOTAIR* also is increasingly recognized as a critical contributor in the metastatic process for a number of cancers more broadly, specifically due to its role as a regulator of epithelial cell plasticity and the epithelial-to-mesenchymal transition (EMT)⁶⁷, although our GSVA analyses did not identify major differences in gene expression in the EMT or other hallmark pathways overall.”

4. In this context, the applied RNAseq data analysis might introduce substantial bias within differential expression analysis and derived results and interpretation. There is no need to deviate from the well-established and accessible methods like DESeq2 and normalization methods that account for e.g. sequencing depth and other technical biases specific to RNAseq data. Details and rationales on e.g. definition of differential gene expression seem arbitrary and are not using commonly accepted thresholds in RNAseq studies. I suggest a reanalysis of the transcriptomic data using state-of-the-art bioinformatic approaches.

Response: We appreciate the opportunity to clarify the methods we used for our RNAseq data. We have added clarification in the Supplementary Methods that:

Pages 5-6: “To evaluate genes differentially expressed between cLNM versus PT samples, we used standardized approaches as described by the Pancancer Analysis of Whole Genomes (PCAWG) Working Group²⁷.”

These methods included normalization based on read counts. We note that we followed the PCAWG methodology rather than DESeq2 because of our large sample size,

according to published literature noting the high false-positive rate of DESeq2 in the population setting when sample sizes are large (PMID: 35292087).

5. The bottom line message of this paper is a higher metastatic potential of fusion drivers in a post-Chernobyl PTC cohort of young patients. I am missing a discussion on potential mechanisms how fusion vs. mutational drivers can impact on the metastatic process. If the paradigm of a multi-step process of tumor progression is true, additional molecular mechanisms must be initiated to enter tumor metastasis. A more detailed discussion on this aspect is needed.

Response: The reviewer raises an important question about the mechanism by which *RET* and other *RTK* fusion drivers have higher metastatic potential compared with other fusion or mutation drivers. We note that *RTKs* influence multiple different signaling pathways, any of which could contribute to the metastatic potential of those tumors, whereas other drivers primarily regulate particular pathways, such as *BRAF* fusion, *BRAF* mutation, and *RAS* mutation specifically acting within the *MAPK* pathway. We have highlighted this potential future research direction with new text in the Discussion:

Page 17: “Future research is needed to understand the higher metastatic potential of *RTKs*, which influence multiple different signaling pathways, in contrast to the lower metastatic potential of other drivers, such as *BRAF* fusion, *BRAF* mutation, and *RAS* mutation, which primarily regulate the *MAPK* pathway specifically.”

*6. Figure 1 depicts a share cLNM yes/no of 40/60 for *BRAF* and 70/30 for *RET* with an OR 3.5 which supports the initial hypothesis. However, the groups Mut (other) vs. Fusion (other) do not reveal differential enrichment of cLNM. Table 1 summarizes the results of multivariate logistic regression on cLNM yes/no separately for *BRAF* and *RET* with mostly insignificant ORs, thus not suggesting differential metastasizing potential. Therefore, additional statistical analysis is needed to include multivariate logistic regression based on the whole data set used for Figure 1 and the covariables of Table 1. Regression analysis should be followed by AUC calculations with and without cross validation (AUC > 0.8 without cross validation and > 0.7 with cross validation are deemed necessary to support initial hypothesis).*

Response: We appreciate the opportunity to clarify the analytic approach and results for our analyses in Figure 1 and Table 1. In Figure 1, we show the raw percentage of cLNM occurrence by driver group. The P-values derive from multivariable models as specified on Page 8:

“...evaluated in sex- and age at PTC-adjusted multivariable models...”

We have added new text to reflect the Reviewer’s observation that the increased frequency of cLNM occurrence in fusion driven tumors did not extend to the “other” category, expanding on the data presented in Table S3 that the 4 tumors in the “other” category with cLNM occurrence all had *BRAF* fusion drivers:

Page 8: “cLNM were most common among tumors with *RET* (N=52/73, 71.2%) or other *RTK* (N=41/64, 64.1%) fusion drivers but less common among tumors with other fusion drivers (N=4/39, 10.3%), all of which occurred in tumors with *BRAF* fusions.”

Table 1, on the other hand, presents results for patient and pathologic characteristics separately for tumors with *BRAF* mutations or *RET* fusions, the two most common groups in our study, again using multivariable models.

We did not perform AUC calculations because we were evaluating associations, rather than constructing a prediction model.

7. Page 16/line 309-314: The reasoning here is somewhat puzzling. The previous Science publication (doi:10.1126/science.abg2538) reports a robust dose response of the driver type marker. In the present study the driver type marker is associated with cLNM prevalence which does not exhibit a dose response. How can both statements be reconciled?

Response: Because occurrence of fusion drivers, age at PTC, and radiation dose are all related to one another, we used multivariable modeling approaches to disentangle the effects of each of these factors on cLNM occurrence. For example, while we observed a non-significant increase in cLNM occurrence with increasing radiation dose in univariate analyses (P=0.21, Table S3), this result was further attenuated in models that adjusted for age at PTC. Age at PTC also appears to be related to cLNM occurrence in univariate analyses (P=4.7x10⁻³), but this effect does not persist in models that control for driver type through stratification. Based on the combined results of our modeling, we concluded that:

Page 15: “Based on multivariable models that included genomic landscape data, our findings suggest that prior reports of strong associations of age at PTC and prior radiation exposure with cLNM occurrence were likely influenced by the relationship of these variables with the PTC driver.”

Page 16: “Notably, prior studies that lacked comprehensive driver data and/or multivariable modeling have reported that younger age at diagnosis, male sex, and exposure to ionizing radiation were associated with increased cLNM occurrence^{4,5,22-27}. However, the increased frequency of fusion drivers among young individuals and those exposed to higher doses of radiation suggest that prior reports associating these characteristics with cLNM occurrence were likely influenced by the relationship of these variables with the PTC driver.”

REVIEWER COMMENTS

Reviewer #1 (Remarks to the Author):

The authors took much effort to thoroughly address my critiques, which is satisfactory.

Reviewer #2 (Remarks to the Author):

In this revised version of the manuscript, the authors have satisfactorily addressed most of the reviewers' concerns raised on their original submission. Although I have no further question, it is relevant to know the histological subtype from both individuals exposed and unexposed, even if it is still not revised according to 2022 WHO classification.

Reviewer #3 (Remarks to the Author):

Morton et al., Genomic characterization of cervical lymph node metastases in papillary thyroid carcinoma following the Chernobyl accident. - NCOMMS-23-06446-A

The authors revised their paper according to my initial comments and criticisms. While I appreciate clarification of several issues and appropriate revision within the manuscript, there are still outstanding issues that need further improvement and clarification:

1. With regard to my initial comment of age-related subgroups of post-Chernobyl PTC, the authors added some marginal explanations in the text for the usage of 30 years as a cut-point (see my initial comment 2/reviewer 3). Arguing with "mean age at PTC diagnosis among exposed individuals" is not appropriate and sufficient since it is common knowledge that morphologic characteristics and aggressiveness of PTC are clearly different between childhood/adolescent PTC and adult PTC. Some co-authors of this manuscript have previously published such findings (see exemplary PMID 15150580, PMID 18651805 and PMID 29989861). In particular, Bogdanova et al. 2018 (PMID 29989861) subdivided their histopathologic analysis of radiogenic and sporadic PTC into children (up to 14 years at PTC), adolescents (15 to 18 years at PTC) and adults (>19 years at PTC) reporting significant differences between age groups. Thus, a cut-point at 19 years discriminating adult and childhood/adolescent PTC would be more appropriate than 30 years used in this study, which generates a subcohort (<30 years) mixed up with young and adult PTC. In this study, 14% are in the childhood/adolescent group vs. 86% in the adult group.

Such a biologically plausible discrimination between young and adult PTC should be made for any comparisons (cLNMs occurrence, dose effect and differential genes expression analyses). I also suspect clear mechanistic explanations if these distinctions are made.

The references mentioned above also suggest a difference for aggressive histopathologic pattern, i.e. solid variants vs. classical variants, and others. I am missing this aspect in the presented study. It is important how these aggressive histologic patterns correlate with the proposed fusion drivers and cLNM enrichment.

2. In response to my initial comment 6) the authors refer to data in Table S3 which should read Table S4 if I understand this correctly. From my point of view there is a problem with the comparison of cLNM enrichment in three different cohorts (primary study population, other Chernobyl cohort, TCGA): while they detect in the primary study cohort a cLNM enrichment for the whole fusion marker group and the subgroup BRAF, RAS, RET, the “other Chernobyl cohort” shows no correlation at all and TCGA only a correlation for BRAF, RAS and RET (but not for the whole fusion marker group). From this observation I would draw the conclusion that a cLNM enrichment in fusion marker PTC can be only seen in their own cohort but not in validation cohorts. The important validation in independent cohorts should be clarified and explicitly explained in the manuscript.

Minor comments:

- Abstract: According to Table S4 percentages and P-values pertain to n=428 cases (and not n=440)
- Table S4: The TCGA cases add up to n=322 and not n=328 as indicated in the header.

Reviewer #2 (Remarks to the Author):

In this revised version of the manuscript, the authors have satisfactorily addressed most of the reviewers concerns raised on their original submission. Although I have no further question, It is relevant to know the histological subtype from both individuals exposed and unexposed, even if is still not revised according to 2022 WHO classification.

Response: The lead pathologist for this study (T. Bogdanova) has an ongoing review of the pathology data for the cases according to the 2022 WHO classification. To address this reviewer comment, we are currently able to provide data for N=174 exposed cases who were diagnosed with PTC prior to age 29 years, and N=81 unexposed cases (all of whom were diagnosed with PTC prior to age 29 years) (see Table below). Cases were classified by dominant growth pattern (papillary, follicular or solid/trabecular) and histological subtype according to the 2022 WHO classification in accordance with review publications on this classification (Baloch et al, 2022; Jung et al, 2022; Basolo et al, 2023), because the 5th edition has not yet been published in its entirety. In this preliminary review, no major differences in the pathology subtypes were found between the groups (see Table). Once the full review of all cases is complete, we will prepare a new manuscript on the relationship between clinical-histopathological characteristics and molecular-genetic alterations in the studied PTCs, thus we have elected to leave this table out of the current manuscript. We have added a statement on the importance of this work to the Discussion:

Page 20: “Future efforts should comprehensively evaluate clinical-histopathological characteristics and molecular-genetic alterations in relation to updated PTC pathologic classifications.”

Reviewer #3 (Remarks to the Author):

Morton et al., Genomic characterization of cervical lymph node metastases in papillary thyroid carcinoma following the Chernobyl accident. - NCOMMS-23-06446-A

The authors revised their paper according to my initial comments and criticisms. While I appreciate clarification of several issues and appropriate revision within the manuscript there are still outstanding issues that need further improvement and clarification:

1. With regard to my initial comment of age-related subgroups of post-Chernobyl PTC the authors added some marginal explanations in the text for the usage of 30 years as a cut-point (see my initial comment 2/reviewer 3). Arguing with “mean age at PTC diagnosis among exposed individuals” is not appropriate and sufficient since it is common knowledge that morphologic characteristics and aggressiveness of PTC are clearly different between childhood/adolescent PTC and adult PTC. Some co-authors of this manuscript have previously published such findings (see exemplary PMID 15150580, PMID 18651805 and PMID 29989861). In particular, Bogdanova et al. 2018 (PMID 29989861) subdivided their histopathologic analysis of radiogenic and sporadic PTC into children (up to 14 years at PTC), adolescents (15 to 18 years at PTC) and adults (>19 years at PTC) reporting significant

differences between age groups. Thus, a cut-point at 19 years discriminating adult and childhood/adolescent PTC would be more appropriate than 30 years used in this study which generates a subcohort (<30 years) mixed up with young and adult PTC. In this study 14% are in the childhood/adolescent group vs. 86% in the adult group.

Such a biologically plausible discrimination between young and adult PTC should be made for any comparisons (cLNM occurrence, dose effect and differential genes expression analyses). I also suspect clear mechanistic explanations if these distinctions are made.

The references mentioned above also suggest a difference for aggressive histopathologic pattern, i.e. solid variants vs. classical variants, and others. I am missing this aspect in the presented study. It is important how these aggressive histologic pattern correlates with the proposed fusion drivers and cLNM enrichment.

Response: The reviewer raises important points regarding two factors: age at PTC diagnosis and pathological characteristics.

Regarding age at PTC, we have included data in Supplementary Table S6 and new text in the Results that specifically highlights the frequency of cLNM occurrence among pediatric cases:

Page 9: “More detailed breakdowns of cLNM occurrence by driver and age at PTC are provided in Table S6. Although results should be interpreted cautiously due to small numbers of cases in certain subgroups, the patterns observed in our overall study population appeared consistent when we restricted to pediatric cases (<20 years), namely a high proportion of cLNM occurrence in PTC with *RET* (76.2%) and other *RTK* (64.3%) fusion drivers and lower percentage for *BRAF* mutations (23.5%).”

We also evaluated our differential expression results restricted to pediatric cases and found similar results. To respond to the reviewer, we added a new figure and the following text to the Results:

Page 15: “Exploratory analyses of the top differential expression results for mRNA (*HOXC10*) and miRNA (miR-196a2) revealed consistent findings when we restricted to pediatric cases (<20 years) (Figure S9).”

We also agree that pathologic characteristics are of great interest. As noted above in our response to Reviewer #2, the lead pathologist for this study (T. Bogdanova) has an ongoing review of the pathology data for the cases according to the 2022 WHO classification. Once the full review of all cases is complete, we will prepare a new manuscript on the relationship between clinical-histopathological characteristics and molecular-genetic alterations in the studied PTCs. We have added a statement on the importance of this work to the Discussion:

Page 20: “Future efforts should comprehensively evaluate clinical-histopathological characteristics and molecular-genetic alterations in relation to updated PTC pathologic classifications.”

2. In response to my initial comment 6) the authors refer to data in Table S3 which should read Table S4 if I understand this correctly. From my point of view there is a problem with the comparison of cLNM enrichment in three different cohorts (primary study population, other Chernobyl cohort, TCGA): while they detect in the primary study cohort a cLNM enrichment for the whole fusion marker group and the subgroup *BRAF*, *RAS*, *RET*, the “other Chernobyl cohort” shows no correlation at all and TCGA only a correlation for *BRAF*, *RAS* and *RET* (but not for the whole fusion marker group). From this observation I would draw the conclusion that a cLNM enrichment in fusion marker PTC can be only seen in their own cohort but not in validation cohorts. The important validation in independent cohorts should be clarified and explicitly explained in the manuscript.

Response: We have further elaborated on the comparison of cLNM enrichment among the three cohorts included in this analysis. In the Results section, we have acknowledged the smaller sample size of the previous Chernobyl studies and added further details regarding the statistical analyses presented in Table S4 that describe the data for the three studies separately:

Page 10: “Although the sample size from the previous Chernobyl studies was limited, the patterns of cLNM occurrence by driver in those studies as well as TCGA generally were consistent with our original study population, though only the differences in cLNM occurrence by specific driver in TCGA were statistically significant (Figure S3, Table S4). Namely, cLNM were more frequent in tumors with fusion than mutation drivers (Chernobyl: 47.9% vs. 30.0%, $P=0.81$; TCGA: 50.9% vs. 44.9%, $P=0.44$), and specifically in tumors with *RET* fusion than *BRAF* mutation drivers (Chernobyl: 55.2% vs. 37.5%, among all specific drivers: $P_{\text{heterogeneity}}=0.39$; TCGA: 78.3% vs. 50.2%, among all specific drivers: $P_{\text{heterogeneity}}=1.4\times 10^{-5}$).”

In the Results section, we also have provided more nuanced interpretation of the pooled analysis results presented in Tables S7-S8:

Page 11: “Similarly, pooled analyses of patient and clinical characteristics demonstrated a consistently increased occurrence of cLNM associated with larger tumor size and more advanced stage for both *BRAF* mutation and *RET* fusion driven tumors among all three studies (Tables S7-S8). In contrast, associations with male sex and younger age in individuals with *BRAF* mutation driven tumors were inconsistent.”

Finally, in the Discussion, we have acknowledged the inter-study differences:

Page 17: “Our findings regarding PTC driver and cLNM frequency were remarkably consistent with those from pediatric PTC studies, although our re-analysis of the previous Chernobyl studies with a younger mean age at PTC found a non-significant increased frequency of cLNM in fusion driven tumors, perhaps due in part to smaller sample size. Nevertheless, our results suggest that patterns of cLNM among young adults are more similar to those of pediatric rather than older adult patients.”

Minor comments:

- Abstract: According to Table S4 percentages and P-values pertain to n=428 cases (and not n=440)

Response: The Abstract has been modified to indicate that the results described included N=428 individuals:

“Among 428 PTC with genomic landscape analyses and known drivers (¹³¹I-exposed=349, unexposed=79; mean age=27.9 years), cLNM were more frequent...”

- Table S4: The TCGA cases add up to n=322 and not n=328 as indicated in the header.

Response: The header has been changed to N=322.

Table. Dominant growth pattern and histological subtypes of ¹³¹I-exposed and unexposed individuals aged <30 years at PTC diagnosis

Parameters	¹³¹ I-exposed n=174		Unexposed n=81	
	N	%	N	%
Dominant growth pattern				
Papillary	91	52.3	47	58.0
Follicular	46	26.4	20	24.7
Solid/trabecular	37	21.3	14	17.3
Histological subtypes (5th Edition, 2022 WHO)*				
Classic PTC	76	43.7	38	46.9
Encapsulated classic PTC	10	5.7	5	6.2
Infiltrative follicular PTC	30	17.2	11	13.6
Solid/trabecular PTC	33	19.0	11	13.6
Warthin-like PTC	5	2.9	5	6.2
Oncocytic PTC	3	1.7	1	1.2
Tall cell PTC	0	-	1	1.2
Encapsulated follicular PTC	16	9.2	9	11.1
Cribiform thyroid carcinoma	1	0.6	0	-

* PTC subtypes registered in these groups

REVIEWERS' COMMENTS

Reviewer #2 (Remarks to the Author):

The authors have addressed/justified all my concerns. Therefore, I consider the manuscript acceptable for publication

Reviewer #3 (Remarks to the Author):

In the revised version of the manuscript the authors have made every effort to take my comments into account. Major concerns have been discussed in this study which mainly confirms earlier results by Franco et al. 2022 (PMID: 35015563). I am still not fully convinced about age-group specific gene expression analyses which could have been presented in more detail. However, the changes made are sufficient to address my initial concerns.